# SWIRL: A Staged Workflow for Interleaved Reinforcement Learning in Mobile GUI Control

## Abstract

The rapid advancement of large vision language models (LVLMs) and agent systems has heightened interest in mobile GUI agents that can reliably translate natural language into interface operations. Existing single-agent approaches, however, remain limited by structural constraints. Although multi-agent systems naturally decouple different competencies, recent progress in multi-agent reinforcement learning (MARL) has often been hindered by inefficiency and remains incompatible with current LVLM architectures. To address these challenges, we introduce SWIRL, a staged workflow for interleaved reinforcement learning designed for multi-agent systems. SWIRL instantiates a round-level training scheme that treats MARL as a sequence of single-agent reinforcement learning subproblems, updating one agent at a time while keeping the others fixed. This formulation enables stable training and promotes efficient coordination across agents. Theoretically, we provide a stepwise safety bound, a cross-round monotonic improvement theorem, and convergence guarantees on return, ensuring robust and principled optimization. In application to mobile GUI control, SWIRL instantiates a Navigator that converts language and screen context into structured plans, and an Interactor that grounds these plans into executable atomic actions. Extensive experiments demonstrate superior performance on both high-level and low-level GUI benchmarks. Beyond GUI tasks, SWIRL also demonstrates strong capability in multi-agent mathematical reasoning, underscoring its potential as a general framework for developing efficient and robust multi-agent systems.

## 1 Introduction

With the rapid progress of large vision–language models (LVLMs) (OpenAI, 2025; Zhu et al., 2025; Bai et al., 2025b; Guo et al., 2025), increasing attention has been devoted to mobile graphical user interface (GUI) agents capable of translating natural language instructions into reliable interface manipulation (Qin et al., 2025; Xu et al., 2024; Wu et al., 2024b; Lu et al., 2024; Luo et al., 2025; Zhang & Zhang, 2024). These agents ground user instructions in the current screenshot and interaction history, reason over this evolving state, and iteratively generate the next action until the task is completed. Effective mobile GUI control depends on two key competencies: task planning, which forms global, goal-conditioned decisions under evolving contexts, and task execution, which translates these plans into executable actions with precise localization. Most existing systems adopt a single-agent design, which complicates the robust integration of these competencies.

We identify two fundamental challenges. First, coupling high-level planning with fine-grained perception and precise actuation makes single end-to-end policies prone to interference and brittleness (Wang et al., 2024a; Erdogan et al., 2025; Mo et al., 2025; Wang et al., 2025). Second, end-to-end systems often exhibit a weak linkage between reasoning traces and executed actions, sometimes producing correct outcomes for spurious reasons or plausible traces paired with faulty actions (Turpin et al., 2023; Arcuschin et al., 2025; Li et al., 2024a), thereby undermining safety and accountability in assurance-critical applications (Zhang et al., 2025; Shi et al., 2025; Kuntz et al., 2025).

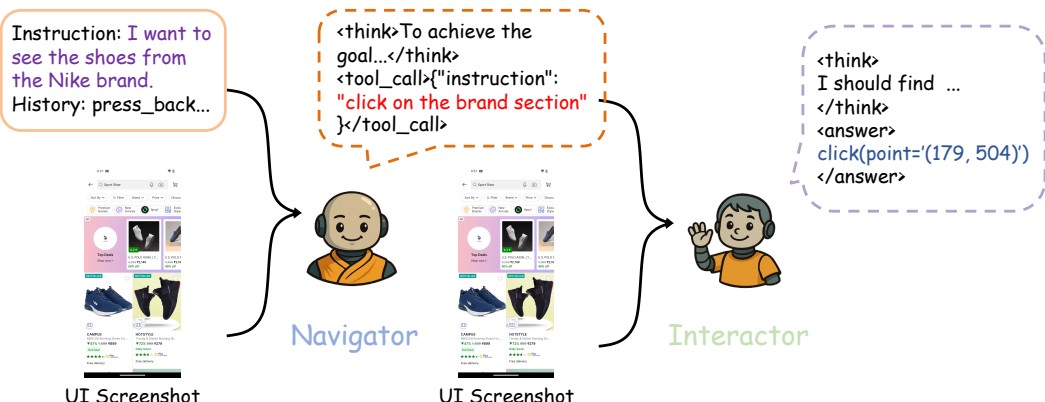

Figure 1: Our multi-agent inference pipelines. Given a high-level instruction, UI screenshots, and historical actions, the Navigator generates a low-level instruction, which the Interactor then uses together with UI screenshots to produce the final executable action. This design decouples task planning from execution, enabling specialization, and leverages explicit intermediate instructions to enhance transparency and interpretability.

A multi-agent design provides a principled approach to decoupling core competencies by assigning planning and execution to specialized agents. Beyond this division of labor, structured interactions between agents further enhance transparency by making the reasoning process more interpretable and the resulting actions more auditable. This explicit linkage between decision-making and execution not only improves accountability but also facilitates supervision and error analysis, which are essential for building reliable GUI control systems. However, training-free adaptation of generic LVLMs rarely suffices for domain-specific GUI control due to insufficient cooperation (Niu et al., 2024). Naive multi-agent reinforcement learning further introduces practical challenges: joint optimization of multiple policies inflates compute budgets and exposes limited capabilities (Wang et al., 2022; Gogineni et al., 2023). Meanwhile, high-throughput reinforcement learning (RL) frameworks developed for LVLMs are almost exclusively engineered for single-agent training (Hu et al., 2024; Sheng et al., 2025), making them ill-suited for MARL (Liao et al., 2025). These limitations motivate a central question: *can we train multi-agent systems that are resource-friendly while ensuring provable stability during training?*

We introduce SWIRL, a staged workflow for interleaved reinforcement learning. SWIRL decomposes multi-agent training into two phases: independent pre-warming of each module, followed by alternating optimization where one module is frozen while the other is updated. During this alternating process, we further incorporate an online reweighting mechanism to enhance training stability and accelerate convergence. SWIRL in Fig. 2(c) updates exactly one agent at a step. After several updates on one agent, SWIRL switches to the next. Unlike traditional concurrent MARL, where all agents are updated simultaneously, SWIRL optimizes the multi-agent objective via round-wise alternating updates: each agent is improved in turn using a standard single-agent RL solver while the other agent's policy is held fixed; over a full round, all agents are updated sequentially. Single-agent RL is used purely as a subproblem solver; the environment and objective remain multi-agent, and all guarantees are stated for the resulting joint policy. Beyond practicality, we offer theoretical and system-level benefits: we establish

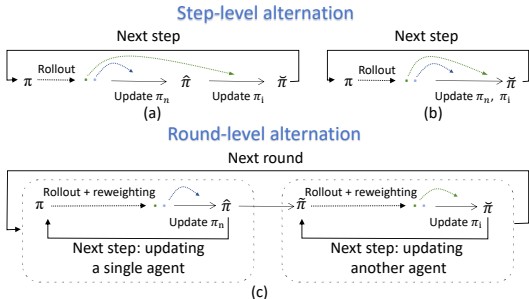

Figure 2: Alternating training paradigm. (a) HAPPO (Zhong et al., 2024): step-level single-agent sequential updates; (b) A2PO (Wang et al., 2023) & MARFT (Liao et al., 2025): enhancing sequential updates with preceding-agent off-policy correction for greater efficiency; (c) SWIRL: round-level alternation with inner solves.

a per-step safety bound, prove monotonic improvement across rounds, and derive a corollary for return convergence. In implementation, SWIRL requires only the currently updated agent to be resident on the training device, yielding $O(1)$ actor memory usage, smooth compatibility with standard stacks, and support for heterogeneous model sizes and update budgets, contrasting with the $O(N)$ memory usage of other methods. Table 1 details the count of actor parameters loaded during training without extra optimizations.

We instantiate SWIRL on mobile GUI control with a dual-agent architecture. The Navigator interprets instructions, interaction history, and the current screenshot to form a task context and produce structured low-level instructions (LLI). The Interactor consumes the LLI together with the current UI view and outputs atomic actions such as click, scroll, and text input with precise localization. Fig. 1 presents the inference pipeline. Training alternates between the two agents: when optimizing the Navigator, the Interactor is fixed and executes the Navigator's instructions to yield actions and rewards; when optimizing the Interactor, the Navigator remains fixed and supplies instructions for each step. This instantiation separates planning from execution and enforces a tight linkage between reasoning and action. With this decoupled design and the stability of interleaved updates, our system attains state-of-the-art zero-shot performance on both high-level and low-level mobile GUI benchmarks using only 3,500 training examples, outperforming baselines trained under diverse supervised fine-tuning (SFT) and RL regimes. We further apply the same interleaved recipe to the mathematics domain and observe significant gains, surpassing those achieved by other multi-agent training approaches.

Table 1: Comparison of actor parameters loaded during training.

| Method | Actor Params |
|---|---|
| HAPPO | $\sum_{i=1}^{N} \lvert\theta_i\rvert$ |
| A2PO | $\sum_{i=1}^{N} \lvert\theta_i\rvert$ |
| MARFT | $\sum_{i=1}^{N} \lvert\theta_i\rvert$ |
| SWIRL (Ours) | $\max\{\lvert\theta_i\rvert\}$ |

The contributions of this paper are as follows: (i) we introduce SWIRL, a multi-agent training framework that interleaves single-agent updates and transforms MARL to a sequence of single-agent RL subproblems; it achieves $O(1)$ actor memory usage by loading only the currently updated agent, and accommodates heterogeneous model sizes, data schedules, and update budgets; (ii) we establish formal guarantees, including a per-step safety bound, a monotonic improvement result across rounds, and convergence of returns; (iii) we instantiate SWIRL for mobile GUI control with a Navigator and an Interactor and, through extensive experiments, show stable training and state-of-the-art zero-shot performance, together with ablations that clarify when interleaving helps; and (iv) we demonstrate transferability by applying the same training recipe to a non-GUI domain (e.g., mathematics) and observe robust improvements on standard benchmarks, indicating potential for broader multi-agent applications.

## 2 METHOD

In Sec. 2.1, we introduce a theoretical multi-agent interleaved updating methodology in Alg. 1 and give the theoretical guarantees. In Sec. 2.2, we formulate the multi-agent framework for GUI navigation. Following that, Sec. 2.3 introduces SWIRL, a practical mobile GUI implementation of the multi-agent interleaved updating method, executed as a two-phase process involving warm-up and round-level alternating RL with online reweighting.

### 2.1 ROUND-LEVEL INTERLEAVED UPDATING

As noted in Sec. 1, our goal is to develop a MARL framework that is both resource-efficient and provably stable. Practically, an ideal approach is round-level interleaved updating: in each round, one agent is updated for several steps while the others remain unchanged, then the process moves to the next agent. This allows us to reuse existing single-agent RL pipelines and nearly single-agent memory cost. However, it raises a key question: *Does this round-level, single-agent–at-a-time update scheme guarantee continual improvement of the overall multi-agent system?*

**ADMM Analogy for Round-Level Interleaving.** A classical algorithm in mathematical optimization offers a useful analogy: the Alternating Direction Method of Multipliers (ADMM) (Boyd et al., 2011). For the constrained problem $\min_{x,y\in\mathbb{R}^n} f(x) + g(y)$ s.t. $Ax + By = c$ and its augmented Lagrangian form: $\mathcal{L}_\rho(x, y, \lambda) = f(x) + g(y) + \lambda^\top(Ax + By - c) + \frac{\rho}{2}\lVert Ax +$

---

**Algorithm 1:** Multi-Agent Interleaved Updating with Monotonic Improvement Guarantee

---

Initialise independent pre-warming $\pi_0^i$, $1 \leq i \leq n$;

**for** *round $k = 0, 1, 2, \ldots$* **do**

    **for** *agent $i = 1, \ldots, n$* **do**

        Initialize $\pi_{k,0}^i \leftarrow \pi_k^i$;

        **for** *step $j = 0, \ldots, K_i - 1$* **do**

            $\pi_{k,j+1}^i \leftarrow \arg\max_{\pi^i} F_{k,i,j}(\pi^i) = \left[ L_{\Pi_{k,i,j}}^i(\tau_k^{-i}, \pi^i) - C_{k,i,j} D_{\mathrm{KL}}^{\max}(\pi_{k,j}^i, \pi^i) \right]$;

        Update $\pi_{k+1}^i \leftarrow \pi_{k,K_i}^i$;

---

$By - c\|^2$. ADMM performs the alternating updates: $x^{k+1} := \arg\min_x \mathcal{L}_\rho(x, y^k, \lambda^k); y^{k+1} := \arg\min_y \mathcal{L}_\rho(x^{k+1}, y, \lambda^k); \lambda^{k+1} := \lambda^k + Ax^{k+1} + By^{k+1} - c$. It decomposes a coupled problem into simpler subproblems while retaining strong convergence guarantees (Glowinski, 2014; Yang et al., 2022). In practice, each subproblem is solved by an inner loop to a prescribed accuracy before the outer iteration advances. The key is that these inner steps deliver enough improvement for the outer objective to make steady progress. Guided by this ADMM viewpoint, our round-level interleaved scheme follows the same inner–outer pattern: as long as the updates of the selected agents in each round guarantee some improvement in the overall joint policy, this alternating scheme remains effective.

**Algorithm and Theoretical Guarantees.** We now formalize this multi-agent round-level interleaved updating training scheme in Alg. 1. The algorithm first independently pre-warms each agent using any single-agent method to obtain initial policies. It then performs interleaved updates: one agent is continuously optimized while the others are fixed, then sequentially switches to the next agent until all are updated, repeating this cycle to decompose MARL into a sequence of single-agent optimization tasks. This structure leads to the following findings: Proposition 1 provides a safety bound for each step, Theorem 1 confirms that the joint return increases consistently across rounds, and Corollary 1 shows the returns converge, and all policy limits attain this value. A summary of notation can be found in Appendix C.1, with complete proofs located in Appendix C.2.

**Proposition 1** (Lower bound at a step). *In round $k$, when agent $\pi_{k,j}^i$ updates to $\pi_{k,j+1}^i$, the new joint policy is $\Pi_{k,i,j+1} := (\tau_k^{-i}, \pi_{k,j+1}^i)$, and the performance satisfies*

$$J(\Pi_{k,i,j+1}) \geq J(\Pi_{k,i,j}) + L_{\Pi_{k,i,j}}^i(\tau_k^{-i}, \pi_{k,j+1}^i) - C_{k,i,j} D_{\mathrm{KL}}^{\max}(\pi_{k,j}^i, \pi_{k,j+1}^i). \tag{1}$$

**Theorem 1** (Monotonic improvement). *Every step updates in Alg. 1 satisfies $F_{k,i,j}(\pi_{k,j+1}^i) \geq 0$, and for all outer rounds $k$ we have $J(\pi_{k+1}) \geq J(\pi_k)$.*

**Corollary 1** (Return convergence). *The sequence $\{J(\pi_k)\}$ approaches a limit, referred to as $\bar{J}$, and the collection of limit points from $\{\pi_k\}$ is non-empty. For any subsequence $\{\pi_{k_j}\}_{j \geq 0}$ that converges such that $\pi_{k_j} \rightarrow \bar{\pi}$, it holds that $J(\bar{\pi}) = \bar{J}$.*

Alg. 1 provides a methodology that is conceptually related to prior alternating-update approaches in MARL (e.g., HAPPO, A2PO, MARFT), but differs in focusing on round-level inner solves and associated theoretical guarantees. In each round, a single active agent performs several steps while all other agents are frozen. Since the baseline stays constant throughout these steps, each step simplifies to a typical single-agent policy update. As a result, the surrogate goal $L - D_{\mathrm{KL}}^{\max}$ can be estimated using well-established single-agent techniques like TRPO (Schulman et al., 2015), PPO (Schulman et al., 2017), and GRPO (Shao et al., 2024), which apply feasible trust region or clipped KL updates. Previous studies have verified both the theoretical validity and empirical effectiveness of this approximation (Schulman et al., 2015; Zhong et al., 2024). We then apply this methodology to the GUI navigation task.

## 2.2 MULTI-AGENT FRAMEWORK FOR GUI CONTROL

**Task Formulation.** We formulate the GUI control task as a sequential decision-making problem. With a natural language instruction $I$, the agent reviews a series of historical screenshots and actions $H_t = \{X_{t-\delta_s}, \ldots, X_{t-1}, r_{t-\delta_a}^a, \ldots, r_{t-1}^a\}$ along with the current screenshot $X_t$ to craft a

structured text reply $r_t$ at each time step $t$. Here, $\delta_s$ and $\delta_a$ denote the counts of past screenshots and actions, respectively. This reply includes a reasoning ($r_t^r$) and a low-level instruction (LLI, $r_t^i$) that outlines the next planned step. The LLI is then succeeded by the actual action ($r_t^a$), where $r_t \sim \pi_{\theta_n, \theta_i}(r \mid I, H_t, X_t)$, $\{r_t^r, r_t^i, r_t^a\} \in r_t$. The objective is to generate the next action $r_t^a$ that complies with instruction $I$. Under the widely adopted offline setting (Li et al., 2024b; Lu et al., 2024), evaluation is single-step and per-state: for each state $(I, H_t, X_t)$ with a human-annotated ground-truth action $\tilde{a}_t$, a prediction $r_t^a$ is correct iff it matches $\tilde{a}_t$ in both action type and parameters (e.g., direction, text, coordinates). Appendix G presents illustrative examples of GUI agents completing GUI control tasks. Nevertheless, navigating GUI-based instructions introduces specific challenges: it necessitates high-level navigation to deduce the next-step instruction and detailed perception to engage with UI components, each demanding distinct skills.

**Architecture and Training Objective.** To proficiently manage the complexities of GUI operations, we utilize a multi-agent system that distinctly separates the responsibilities between the *Navigator* ($\pi_{\theta_n}$) and the *Interactor* ($\pi_{\theta_i}$). The *Navigator* is responsible for high-level planning, where it interprets the natural language instructions, merges past actions with the UI views, and establishes a coherent task context with reasoning. It then creates a detailed LLI, reflecting the intended next actions, based on the reasoning process. Subsequently, the *Interactor* combines the LLI and the current UI view to generate concrete atomic actions, including actions like click, scroll, etc. This involves precise cursor positioning and visual interpretation to ensure accurate execution of the planned steps within the interface. The inference pipeline of the system is illustrated in Fig. 1.

Building on the system architecture above, we train the two agents with a round-level interleaved scheme (Alg. 1). In each round, we select one role, either the Navigator or the Interactor, and run multiple inner updates on its parameters while freezing the other agent. We then swap roles and repeat. Optimizing the theoretical update for an individual agent demands the calculation of advantage $A$, surrogate $L$, and $D_{\mathrm{KL}}^{\max}$, which is cost-prohibitive. Consequently, we implement practical relaxations, similar to (Zhong et al., 2024), by approximating the theoretical goal using a GRPO objective (Shao et al., 2024). Concretely, we calculate single-agent improvements using group-relative advantages, denoted as $A_k$, which are derived from multiple rollouts (standardized across the batch). We control $D_{\mathrm{KL}}^{\max}$ with two tractable terms: clipped ratios around $\pi_{\mathrm{old}}$ and a KL anchor to $\pi_{\mathrm{ref}}$, ensuring local trust-region control and curbing drift for stability. This preserves the ascent direction of the theoretical target $L - D_{\mathrm{KL}}^{\max}$, whilst ensuring stability and efficiency. Each agent's action is an autoregressive sequence, responses for rollouts $K$ are $\{r_{k,\ell}\}_{1 \le k \le K}^{1 \le \ell \le |r_k|}$. We compute token-wise importance ratios aligned with GRPO, in line with (Luo et al., 2025). Each token is assigned to either the navigator or interactor by the indicator $\mathbb{I}_{k,\ell}^{(j)} = 1$ if and only if $r_{k,\ell} \sim \pi_{\theta_j}$ (and 0 otherwise), securing agent-specific credit assignment without breaching the "freeze-the-complement" rule. In conclusion, our overarching multi-agent training objective is:

$$\mathcal{J}(\theta_n, \theta_i) = \mathbb{E}_{(I, H_t, X_t) \sim \mathcal{D}}^{r \sim \pi} \left[ \sum_{k,\ell} \sum_{j \in \{n,i\}} \frac{\mathbb{I}_{k,\ell}^{(j)}}{K \sum_{\ell=1}^{|r_k|} \mathbb{I}_{k,\ell}^{(j)}} \cdot (\mathrm{clsu}(v_{k,\ell}^{(j)}, A_k) - \lambda D_{\mathrm{KL}}[\pi_{\theta_j} \| \pi_{\theta_j}^{\mathrm{ref}}]) \right]. \quad (2)$$

The clipped surrogate is: $\mathrm{clsu}(v, A) = \min\left(vA, \mathrm{clip}(v, 1-\epsilon, 1+\epsilon)A\right)$, where the value of importance ratio is: $v_{k,\ell}^{(j)} = \frac{\pi_{\theta_j}(r_{k,\ell}|I, H_t, X_t, r_{k,<\ell})}{\pi_{\theta_j}^{\mathrm{old}}(r_{k,\ell}|I, H_t, X_t, r_{k,<\ell})}$. The scalar reward is composed of: $R_k = \alpha R_{\mathrm{form}} + \beta R_{\mathrm{acc}}$, where $R_{\mathrm{form}}$ denotes the reward for format correctness (e.g., proper usage of required tags), and $R_{\mathrm{acc}}$ is the reward for action accuracy, defined as $R_{\mathrm{acc}} = \lambda_1 R_{\mathrm{act}} + \lambda_2 R_{\mathrm{info}}$, where $R_{\mathrm{act}}$ measures the correctness of the predicted action type, and $R_{\mathrm{info}}$ quantifies the accuracy of the action parameters (e.g., the click location). Finally, the normalized advantage is computed as $A_k = \frac{R_k - \mu}{\sigma}$, where $\mu$ and $\sigma$ are the mean and standard deviation of rewards across sampled trajectories.

## 2.3 SWIRL: Staged Workflow for Interleaved Reinforcement Learning

Building on the multi-agent architecture and learning objective described above, we introduce **SWIRL**, a Staged Workflow for Interleaved Reinforcement Learning. This approach is crafted to efficiently coordinate and enhance the performance of the Navigator and Interactor through round-

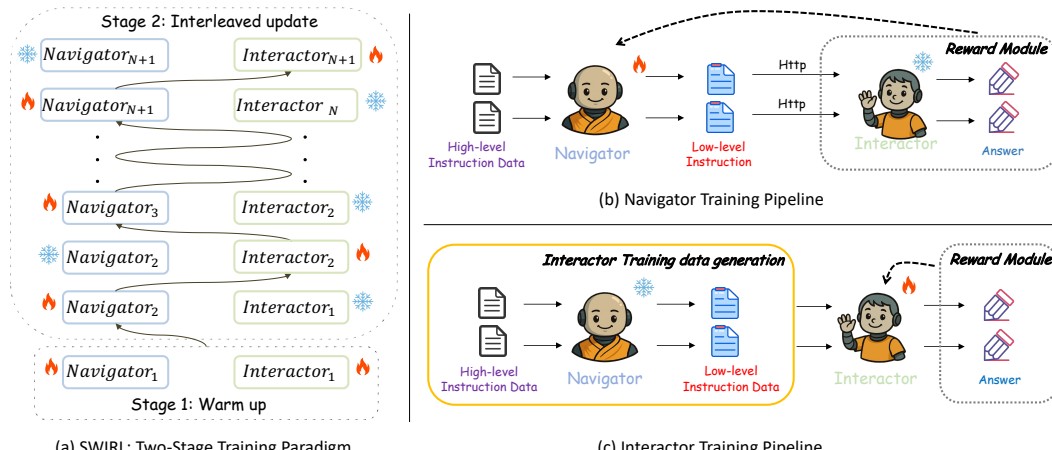

Figure 3: Our multi-agent training pipeline. (a) SWIRL decomposes multi-agent learning into two stages. Stage 1 performs warm-up, where each agent module (Navigator and Interactor) is initialized independently. Stage 2 proceeds with interleaved updates, where optimization alternates between agents: one module is updated while the other is frozen. (b) Given high-level instruction training data, the Navigator generates low-level instructions and obtains rewards by making HTTP calls to a reward module that includes the Interactor (indicated by the dashed box). (c) First, the Navigator generates training data (low-level instructions) for the Interactor (indicated by the orange box), which are then used to train the Interactor.

, as demonstrated in Fig. 3 and Alg. 2. The benefits of SWIRL are listed in Appendix. D.3.

**Stage 1: Warm-up initialization.** The Navigator is first initialized through lightweight Chain-of-Thought (Wei et al., 2022) SFT, while the interactor is bootstrapped via initial reinforcement learning. The primary goal of this stage is to let each agent clearly learn its designated role: the navigator focuses on producing reasoning steps and LLI, whereas the interactor outputs the corresponding action. This warm-up phase establishes a robust foundation and minimizes variability at the outset.

**Stage 2: Interleaved update.** After warm-up, SWIRL progresses into a round-level alternating training stage. Within each round, one agent undergoes continuous optimization via reinforcement learning, while the other is kept static. This approach simplifies the intertwined multi-agent learning challenge into a series of static single-agent tasks. Therefore, it allows us to directly leverage modern single-agent RL algorithms, such as GRPO (see Alg. 3), thereby decreasing implementation complexity while preserving training stability and cooperative efficiency. In the Navigator training process, the Navigator generates a reply $r = \{r^r, r^i\}$ at each step. The frozen Interactor then executes $r^i$ to produce the final action $r^a$. Rewards are computed as described in Sec. 2.2, with the accuracy reward of $r^a$ serving as the Navigator's $R_{\text{acc}}$. To further enhance training efficiency and scalability, we deploy the Interactor on a separate server and integrate it within the reward module. The Navigator communicates with the Interactor via HTTP requests, enabling efficient reward computation during RL optimization. For Interactor training, we first use the frozen Navigator to generate low-level instructions for each training sample. The Interactor is then optimized via RL using these instructions as input, with rewards also computed according to Sec. 2.2. The training pipelines for the Navigator and Interactor are illustrated in Fig. 3b and Fig. 3c, respectively.

**Online reweighting.** After computing the GRPO-type advantages for each batch, we exclude those low-quality samples (Meng et al., 2025; Cui et al., 2025) by regulation $\mathcal{R}$. These low-quality samples typically arise due to collaborator mistakes, noise, or simplicity (Shi et al., 2025). As the model's performance increases, it might happen that the number of high-quality samples meeting our filtering standard becomes fewer than the batch size, leading to partially filled batches. To address this, we replenish the batch by randomly resampling from the remaining high-confidence instructions. This process produces reliable batches without introducing additional rollout cost, implicitly up-

weights the informative samples to enhance training stability and convergence. Details are provided in Appendix D.2.

---

**Algorithm 2:** SWIRL: Staged Workflow for Interleaved Reinforcement Learning

---

**Input:** Dataset $\mathcal{D}$, hyperparameters $N, \mu_n, \mu_i$;

**Stage 1: Warm-up initialization.**

Initialize navigator $\pi_{\theta_n}^{(1)}$ via supervised fine-tuning on instruction-action pairs;

Initialize interactor $\pi_{\theta_i}^{(1)}$ via RL using fixed planner outputs;

**Stage 2: Interleaved reinforcement learning.**

**for** $k = 1$ **to** $N$ **do**

> **Navigator update:** freeze $\pi_{\theta_i}^{(k)}$, update $\pi_{\theta_n}$ using RL in Alg. 3:
>
> $$\theta_n^{(k+1)} \leftarrow RL(\theta_n \mid \mathcal{J}, \theta_n^{(k)}, \theta_i^{(k)}, \mu_n);$$
>
> **Interactor update:** freeze $\pi_{\theta_n}^{(k+1)}$, update $\pi_{\theta_i}$ using RL in Alg. 3:
>
> $$\theta_i^{(k+1)} \leftarrow RL(\theta_i \mid \mathcal{J}, \theta_n^{(k+1)}, \theta_i^{(k)}, \mu_i);$$

---

# 3 EXPERIMENT

The experimental setup is described in Sec. 3.1, and the full experimental details are provided in Appendix E. Sec. 3.2 and Sec. 3.3 present evaluations of SWIRL's zero-shot performance on high-level and low-level tasks, respectively. Sec. 3.4 investigates the proposed multi-agent training framework in the mathematics domain. The ablation study is presented in Sec. 3.5, with additional experiments reported in Appendix F.

## 3.1 SETTINGS

**Implementation Details.** We use Qwen2.5-VL-3B (Bai et al., 2025b) as the base model for both the Navigator and Interactor agents. For historical context, we include only the sequence of past actions, excluding previous screenshots (i.e., $\delta_a = t - 1$, $\delta_s = 0$). The weighting coefficients in the reward function in Sec. 2.2 are set to $\alpha = 0.1$, $\beta = 0.9$, $\lambda_1 = 0.2$, and $\lambda_2 = 0.8$. For online reweighting, we define the rule $\mathcal{R}$ as: $\mathcal{R} = \begin{cases} \text{retain}, & 0.1 < \overline{R_x} < 1, \\ \text{discard}, & \text{otherwise.} \end{cases}$, where $\overline{R_x}$ denotes the average reward of sample $x$ across rollouts.

**Training.** We collected 1,500 and 2,000 mobile control samples for Stage 1 and Stage 2, respectively (see Appendix E.1 for details). In Stage 1 (warm-up), the Navigator is trained for 1 epoch with SFT and the Interactor for 5 epochs with GRPO (Shao et al., 2024), both using a learning rate of $1 \times 10^{-6}$, batch size 32, and DeepSpeed ZeRO-1 for the Navigator; the Interactor uses 5 rollouts per sample. In Stage 2 (SWIRL alternating updates), the two agents are alternately trained for 2 epochs each per round over 20 rounds, maintaining the same hyperparameters; the Navigator uses 8 rollouts per sample and the Interactor 5. Stage 1 runs on $8\times$ NVIDIA A800 GPUs; Stage 2 uses $16\times$ A800s, with half for Interactor deployment as a vLLM-based (Kwon et al., 2023) inference service and half for training. The training framework builds on Qwen2.5-VL[1] (Bai et al., 2025b) for SFT and VeRL[2] (Sheng et al., 2025) for RL.

**Evaluation.** We evaluate SWIRL on high-level tasks (AndroidControl-High (Li et al., 2024b), GUIOdyssey (Lu et al., 2024)) and low-level tasks (AndroidControl-Low (Li et al., 2024b), GUI-Act (Chen et al., 2024), OmniAct (Kapoor et al., 2024)). Following the previous work (Wu et al.,

---

[1] https://github.com/QwenLM/Qwen2.5-VL/tree/main/qwen-vl-finetune

[2] https://github.com/volcengine/verl

Table 2: Performance comparison on the AndroidControl-High and GUIOdyssey datasets. **Bold** and underline indicate the best and second-best performers, respectively.

| Models | Method | AndroidControl-High | | | GUIOdyssey | | | Overall |
| | | Type | GR | SR | Type | GR | SR | |
|---|---|---|---|---|---|---|---|---|
| GPT-4o | Closed-Source | 63.06 | 30.90 | 21.17 | 37.50 | 14.17 | 5.36 | 28.69 |
| OS-Atlas-4B | SFT | 49.01 | 49.51 | 22.77 | 49.63 | 34.63 | 20.25 | 37.63 |
| OS-Atlas-7B | SFT | 57.44 | 54.90 | 29.83 | 60.42 | 39.74 | 26.96 | 44.88 |
| UI-R1-3B | RFT | 43.97 | 63.99 | 26.30 | 20.93 | 56.35 | 8.65 | 36.70 |
| UI-R1-E-3B | RFT | 29.67 | 61.50 | 14.37 | 7.59 | 62.87 | 1.94 | 29.66 |
| GUI-R1-3B | RFT | 58.04 | 56.24 | 46.55 | 54.84 | 41.52 | 41.33 | 49.75 |
| GUI-R1-7B | RFT | **71.63** | 65.56 | **51.67** | 65.49 | 43.64 | 38.79 | 56.13 |
| ReGUIDE-7B | RFT | – | – | 50.00 | – | – | – | – |
| GPT-5 / UI-R1-3B | Multi-Agent | 55.08 | 73.47 | 37.27 | 62.97 | 62.42 | 35.93 | 54.52 |
| GPT-5 / UI-R1-E-3B | Multi-Agent | 60.36 | **74.79** | 40.65 | 65.28 | 63.41 | 38.22 | 57.12 |
| GPT-5 / Interactor | Multi-Agent | 64.68 | 74.62 | 49.53 | 68.77 | 62.70 | 44.21 | 60.75 |
| Navigator / Interactor | SWIRL | 66.72 | 71.19 | 51.24 | **74.87** | **66.39** | **51.65** | **63.68** |

Table 3: Performance comparison on web and desktop low-level tasks. The best and second-best in each column are indicated by **bold** and underline, respectively.

| Models | GUI-Act-Web | | | OmniAct-Web | | | OmniAct-Desktop | | | Overall |
| | Type | GR | SR | Type | GR | SR | Type | GR | SR | |
|---|---|---|---|---|---|---|---|---|---|---|---|
| GPT-4o | 77.09 | 45.02 | 41.84 | 79.33 | 42.79 | 34.06 | 79.97 | 63.25 | 50.67 | 57.11 |
| OS-Atlas-4B | 79.22 | 58.57 | 42.62 | 46.74 | 49.24 | 22.99 | 63.30 | 42.55 | 26.94 | 48.02 |
| OS-Atlas-7B | 86.95 | 75.61 | 57.02 | 85.63 | 69.35 | 59.15 | 90.24 | 62.87 | 56.73 | 71.51 |
| UI-R1-3B | 80.69 | 80.07 | 67.92 | 75.42 | 61.35 | 61.33 | 73.41 | 64.12 | 63.98 | 69.81 |
| GUI-R1-3B | 94.07 | 86.86 | 73.53 | 88.58 | 75.10 | 75.08 | 91.86 | 78.37 | 78.31 | 82.42 |
| GUI-R1-7B | 93.15 | **88.33** | 76.15 | 91.16 | 77.29 | **77.35** | 92.20 | **83.36** | **83.33** | 84.70 |
| Interactor | **95.00** | 87.85 | **84.85** | **94.52** | **81.67** | 77.32 | **94.65** | 77.09 | 72.97 | **85.10** |

Table 4: Math reasoning results with 7B models. SOLO denotes the untrained base model, while ReMA and SWIRL apply multi-agent training on the same base. Parentheses indicate improvements/degradations over SOLO.

| Method | MATH500 | GSM8K | Minerva | OlympiadBench | Gaokao2023en | AMC23 | AIME24 | Overall |
|---|---|---|---|---|---|---|---|---|
| SOLO | **75.0** | **92.0** | 35.7 | 38.2 | 56.6 | 47.5 | 6.7 | 50.2 |
| ReMA | 74.4 (-0.6) | 90.6 (-1.4) | 34.9 (-0.8) | 36.3 (-1.9) | 57.9 (+1.3) | **57.5 (+10.0)** | **20.0 (+13.3)** | 53.1 (+2.9) |
| **SWIRL** | 74.4 (-0.6) | 91.9 (-0.1) | **36.0 (+0.3)** | **41.9 (+3.7)** | **64.9 (+8.3)** | **57.5 (+10.0)** | 10.0 (+3.3) | **53.8 (+3.6)** |

2024b; Luo et al., 2025), we report Type, GR, and SR in a zero-shot prompt setting to assess out-of-domain generalization. Appendix E.2 provides detailed information. All evaluations are conducted in a zero-shot prompt setting to assess the models' out-of-domain generalization ability.

## 3.2 HIGH-LEVEL TASK ZERO-SHOT PERFORMANCE

We benchmark our approach against models trained under different paradigms and observe substantial gains. As shown in Table 2, our multi-agent framework with two 3B models achieves state-of-the-art zero-shot performance, surpassing the SFT-trained OS-Atlas-7B (Wu et al., 2024b) by 18.8 points and the RFT-trained GUI-R1-7B (Luo et al., 2025) by 7.55 points. When GPT-5 (OpenAI, 2025) is used as a high-level to low-level planner for UI-R1-3B and UI-R1-E-3B (Lu et al., 2025), the scores increase by 17.82 and 24.86 points, confirming the advantage of separating planning from execution. Replacing GPT-5 with our Navigator trained using SWIRL and interleaved updates yields a further improvement of 2.93 points, demonstrating the effectiveness of our training strategy in enhancing coordination and boosting downstream performance.

## 3.3 Low-Level Task Zero-shot Performance

The Interactor trained with SWIRL interleaved updates can also operate independently as a low-level task executor. Although its low-level instruction inputs during Stage 2 training are generated by the Navigator, it still achieves strong results, recording the highest SR and GR scores (78.81 and 92.20) and the second-highest Type score (84.62) on the AndroidControl-Low benchmark (Li et al., 2024b), as shown in Table 5. Notably, despite all 3,500 training samples across both stages originating exclusively from mobile devices, the model achieves an overall score of 85.10 on low-level tasks in the web- and desktop-based GUI-Act (Chen et al., 2024) and Omni-Act (Kapoor et al., 2024) datasets, representing the best average performance (Table 3). This remarkable cross-domain generalization underscores the robustness of our model and provides strong empirical evidence for the effectiveness of our training methodology.

Table 5: AndroidControl-Low Results. **Bold** and underline indicate the best and second-best performers, respectively.

| Model | AndroidControl-Low | | |
| --- | --- | --- | --- |
| | Type | GR | SR |
| GPT-4o | 74.33 | 38.67 | 28.39 |
| OS-Atlas-4B | 64.58 | 71.19 | 40.62 |
| OS-Atlas-7B | 73.00 | 73.37 | 50.94 |
| UI-R1 | 72.49 | 88.48 | 57.37 |
| UI-R1-E | 73.91 | 91.91 | 55.58 |
| GUI-R1-3B | 83.58 | 81.59 | 64.41 |
| GUI-R1-7B | **85.17** | 84.02 | 66.52 |
| ReGUIDE-7B | – | – | 67.40 |
| SE-GUI-7B | – | 79.60 | 68.20 |
| Interactor | 84.62 | **92.20** | **78.81** |

## 3.4 Multi-Agent Training Framework in the Mathematics Domain

We adapt SWIRL to the domain of mathematical reasoning to assess the generalizability and transferability of our multi-agent training framework across models of different scales. For fair comparison, we follow prior work and adopt the same base models and data: ReMA (Wan et al., 2025) uses Qwen2.5-7B-Instruct (Team, 2024), MARFT (Liao et al., 2025) uses Qwen2.5-Coder-3B-Instruct (Hui et al., 2024), and both are trained on the MATH training set (Hendrycks et al., 2021). Implementation details are provided in Appendix E.3. Results with 7B models are presented in Table 4. SWIRL achieves

Table 6: Math reasoning results with 3B models. SOLO denotes the untrained single model, while MARFT and SWIRL apply multi-agent training on the same base.

| Method | MATH500 | CMATH | GSM8K | Overall |
| --- | --- | --- | --- | --- |
| SOLO | 40.4 | 81.4 | 76.8 | 66.2 |
| MARFT | 49.8 | 83.0 | 78.7 | 70.5 |
| **SWIRL** | **64.6** | **83.5** | **81.4** | **76.5** |

the highest overall score of 53.8, yielding larger gains over the single-model baseline across most benchmarks. Results with 3B models are shown in Table 6, where SWIRL consistently improves performance on all benchmarks, delivering an average gain of 10.2 points over the base model. These findings demonstrate that SWIRL not only excels in its original GUI mobile control domain but also transfers effectively to a distinct problem setting, underscoring its robustness and broad applicability.

## 3.5 Ablation Study

**Effect of Interleaved Update.** SWIRL's second stage employs interleaved updates to sequentially optimize all agents within the multi-agent system. To contextualize the benefit of this design, we additionally consider an alternative strategy that performs isolated training: the Navigator is trained to convergence first, and then the Interactor is trained while keeping the well-performing Navigator frozen. Notably, this isolated strategy can be viewed as a degenerate case of interleaved updates, corresponding to using only one round of updates. To compare these training strategies, we train with the same 2,000 Stage-2 samples under these two settings: the proposed interleaved updates and the isolated update scheme. For isolated updates, we evaluate both RL-based training and supervised fine-tuning (SFT). As shown in Table 7, isolated updates using either SFT or RL yield modest improvements (+1.68 and +1.47, respectively). In contrast, interleaved updates produce a substantially larger gain (+3.67), demonstrating the clear advantage of iteratively coordinating the agents rather than optimizing them in isolation.

**Further Experiments.** Additional experiments are provided in Appendix F, including analyses of SWIRL's impact on individual agents, investigations into training stability, and detailed ablations of key mechanisms such as the effect of interleaving frequency and the online reweighting strategy.

Table 7: Effect of interleaved updates and isolated updates. Numbers in *italics* show performance gains relative to the baseline.

| Training Strategy | AndroidControl-High | | | GUIOdyssey | | | Overall |
|---|---|---|---|---|---|---|---|
| | Type | GR | SR | Type | GR | SR | |
| Baseline | 62.87 | 69.00 | 46.83 | 70.73 | 63.32 | 46.42 | 60.01 |
| + Isolated SFT | 63.77 *(+0.90)* | 70.25 *(+0.35)* | 48.04 *(+1.21)* | 72.25 *(+1.52)* | 66.19 *(+2.87)* | 49.64 *(+3.22)* | 61.69 *(+1.68)* |
| + Isolated RL | 64.87 *(+2.00)* | 70.68 *(+1.68)* | 49.34 *(+2.51)* | 72.01 *(+1.28)* | 63.98 *(+0.66)* | 47.97 *(+1.55)* | 61.48 *(+1.47)* |
| + Interleaved Learning | **66.72** *(+3.85)* | **71.19** *(+1.29)* | **51.24** *(+4.41)* | **74.87** *(+4.14)* | **66.39** *(+3.07)* | **51.65** *(+5.23)* | **63.68** *(+3.67)* |

## 4 CONCLUSION

In this paper, we presented SWIRL, an interleaved reinforcement learning paradigm that reformulates multi-agent training into tractable single-agent updates. The central principle of SWIRL lies in its round-level alternating training strategy, where in each round one agent is continuously optimized through reinforcement learning while the others remain fixed. We provided theoretical guarantees for stable optimization and validated the effectiveness of SWIRL through extensive experiments in both mobile GUI control and multi-agent mathematical reasoning. Looking forward, we hope that SWIRL can inspire new approaches to multi-agent training in broader domains, such as finance and AI for science, where efficient coordination and reliable optimization remain critical challenges.

## ETHICS AND REPRODUCIBILITY STATEMENT

All data used in our experiments are obtained from previously released and widely adopted datasets, with their sources clearly documented and properly cited. All open-source libraries and resources employed in this study are also fully specified. As such, this study does not raise concerns related to discrimination, bias, or fairness. Furthermore, our models are not expected to generate harmful content. To ensure reproducibility, we provide detailed descriptions of the experimental setup in Sec. 3.1 and additional implementation details in Appendix E. We also provide the complete source code directly in the supplementary material, while anonymized links to the datasets and model checkpoints are included to facilitate replication.

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

APPENDIX

## A    LLM USAGE

We used large language models (LLMs) only as auxiliary tools for grammar checking, language polishing, and logo generation. All outputs were carefully reviewed by the authors, who take full responsibility for the final manuscript.

## B    RELATED WORK

**Reinforcement Learning for GUI Control.**    Reinforcement learning (RL) has recently emerged as a promising paradigm for GUI tasks. Unlike supervised fine-tuning (SFT), which requires large-scale annotated data, RL can learn effective policies from comparatively fewer samples while exhibiting stronger generalization to new tasks (Chu et al., 2025). Several recent methods use RL to train and evaluate the GUI agent in dynamic environment settings (Bai et al., 2024; 2025a; Qi et al., 2024; Wu et al., 2025; Wang et al., 2024b). Meanwhile, another research direction has explored RL for GUI grounding using static offline data, with research directions ranging from reward function design to policy optimization (Lu et al., 2025; Luo et al., 2025; Liu et al., 2025; Zhou et al., 2025; Yuan et al., 2025; Tang et al., 2025; Lee et al., 2025). Furthermore, existing work has predominantly adopted single-agent settings, leaving multi-agent approaches largely unexplored.

**Multi-Agent Systems Based on Large Language Models.**    A parallel line of research explores multi-agent systems powered by LLMs to address complex tasks (Hu et al., 2025; Xiao et al., 2024; Xiang et al., 2025; Zhao et al., 2024b; Wu et al., 2024a; Zhao et al., 2024a; Du et al., 2023; Wu et al., 2025). These systems typically assign specialized roles such as debating, voting, negotiation, thereby structuring interactions and facilitating coordination without training. To enhance robustness and mitigate distribution shift (Han et al., 2024), recent studies have proposed strategies including persuasion-aware training (Stengel-Eskin et al., 2024) and iterative self-improvement via SFT (Subramaniam et al., 2025; Zhao et al., 2025). An important challenge is how to effectively train model cooperation when only limited training data is available (Tran et al., 2025).

**Multi-Agent Reinforcement Learning.**    Works in Multi-Agent Reinforcement Learning (MARL) largely falls into value-based methods (Sunehag et al., 2017; Rashid et al., 2020) and actor-critic approaches (Chu et al., 2019; De Witt et al., 2020). A core challenge is non-stationarity, as one agent's update changes others' observations. Alternating optimization (e.g., A2PO (Wang et al., 2023), HARL (Zhong et al., 2024)) update agents sequentially at the step level but still face scalability issues (Canese et al., 2021; Tran et al., 2025). ReMA (Wan et al., 2025) proposes a fixed dual-agent paradigm that separates meta-thinking and reasoning, sequentially optimized with multi-agent reinforcement learning. MARFT (Liao et al., 2025) combines MARL with LLMs for mathematical problem but suffers from gradient conflicts and parameter drift as the scale increases. It further argues that unifying MARL and LLMs is harder than addressing either alone, highlighting the need for scalable frameworks to integrate them efficiently.

## C    PRELIMINARY AND PROOFS

### C.1    PRELIMINARY WITH NOTATION

**Environment and policies.**    We consider a cooperative Markov game $\langle \mathcal{N}, \mathcal{S}, \mathcal{A}, r, P, d \rangle$ with $n$ agents: $\mathcal{N} = [n] = \{1, \ldots, n\}$. $\mathcal{S}$ is the state space. Let $\mathcal{A} := \prod_{i \in \mathcal{N}} \mathcal{A}^i$ be the joint action space. For each $i \in \mathcal{N}$, the (stochastic Markov) policy $\pi^i(\cdot \mid s) \in \Delta(\mathcal{A}^i)$ together form the joint policy $\pi = (\pi^i)_{i \in \mathcal{N}}$, which induces the joint action distribution $\pi(a \mid s) = \prod_{i \in \mathcal{N}} \pi^i(a^i \mid s)$ for $a = (a^i)_{i \in \mathcal{N}}$. Here $\Delta(\cdot)$ denotes the set of probability distributions over a set. The environment transitions according to $P : \mathcal{S} \times \mathcal{A} \to \Delta(\mathcal{S})$, written $s_{t+1} \sim P(\cdot \mid s_t, a_t)$, with initial state $s_0 \sim d$ and joint reward $r : \mathcal{S} \times \mathcal{A} \to \mathbb{R}$. Let $\rho_\pi$ be the state-visitation distribution induced by the joint policy $\pi$. For agent-wise decomposition we can write $a = (a^{-i}, a^i)$ and denote $\pi^{-i} = (\pi^j)_{j \neq i}$.

**Joint return $J(\pi)$.** For discount $\gamma \in [0, 1)$ and any fixed initial-state distribution,

$$J(\pi) := \mathbb{E}_\pi \left[ \sum_{t \geq 0} \gamma^t \, r(s_t, a_t) \right].$$

It represents the expected total reward from this point forward under the policy $\pi$: summing the reward from each subsequent step and applying a discount factor of $\gamma^t$ at step $t$, where $s_0 \sim d, a_t \sim \pi(\cdot \mid s_t), s_{t+1} \sim P(\cdot \mid s_t, a_t)$. In tasks with a finite horizon, this sum naturally concludes upon task completion. The discount factor $\gamma$ dictates the temporal range: $\gamma = 0$ focuses only on immediate rewards, suitable for offline tasks, where $J(\pi) = \mathbb{E}_\pi[r(s_0, a_0)]$, whereas a higher value of $\gamma$ emphasizes rewards further in the future.

**Rounds, order, and steps.** Outer rounds are indexed by $k = 0, 1, 2, \ldots$. Agent $i$ executes a block of $K_i$ *steps* indexed by $j = 0, \ldots, K_i - 1$, and we denote a step by $(k, i, j)$. The joint policy at the start (resp. end) of round $k$ is $\pi_k$ (resp. $\pi_{k+1}$). During agent $i$'s block, $\pi_{k,j}^i$ is its temporary policy after $j$ steps, initialized by $\pi_{k,0}^i = \pi_k^i$; after finishing the block, set $\pi_{k+1}^i := \pi_{k,K_i}^i$. Other agents are held fixed according to the rolling baseline defined next.

**Rolling baseline and complement policy.** The baseline joint policy at step $(k, i, j)$ is

$$\Pi_{k,i,j} := \big( \{\pi_{k+1}^r\}_{r<i}, \, \pi_{k,j}^i, \, \{\pi_k^r\}_{r>i} \big) = \big( \tau_k^{-i}, \, \pi_{k,j}^i \big),$$

where the complement policy (all agents except $i$) is

$$\tau_k^{-i} := \big( \{\pi_{k+1}^r\}_{r<i}, \, \{\pi_k^r\}_{r>i} \big).$$

During step $j$ of agent $i$ in the $k$th round, agents positioned earlier in the sequence ($r < i$) have already adopted their policies for round $(k+1)$, the current agent follows its internal iterate $\pi_{k,j}^i$, and those positioned later ($r > i$) continue to use their round-$k$ policies.

**Value, $Q$, and advantages.** Define

$$V_\pi(s) := \mathbb{E}_\pi \Big[ \sum_{t \geq 0} \gamma^t r(s_t, a_t) \,\Big|\, s_0 = s \Big], \qquad Q_\pi(s, a) := \mathbb{E}_\pi \Big[ \sum_{t \geq 0} \gamma^t r(s_t, a_t) \,\Big|\, s_0 = s, \, a_0 = a \Big].$$

The advantage is $A_\pi(s, a) := Q_\pi(s, a) - V_\pi(s)$.

For agent $i$, define the marginal state–action value and the single-agent advantage by

$$Q_\pi^i(s, a^i) := \mathbb{E}_{a^{-i} \sim \pi^{-i}(\cdot|s)} \big[ Q_\pi\big(s, (a^{-i}, a^i)\big) \big], \qquad A_\pi^i(s, a^i) := Q_\pi^i(s, a^i) - V_\pi(s).$$

$A_\pi^i(s, a^i)$ is the local improvement if agent $i$ takes $a^i$ at $s$ while others act according to $\pi^{-i}$. A basic identity we use is the zero-mean property $\mathbb{E}_{a^i \sim \pi^i(\cdot|s)} \big[ A_\pi^i(s, a^i) \big] = 0$ for every $s$ as stated in (Zhong et al., 2024).

**Surrogate improvement.** Given the baseline $\Pi_{k,i,j}$, the complement $\tau_k^{-i}$, and a candidate policy $\hat{\pi}^i$ for agent $i$, define

$$L_{\Pi_{k,i,j}}^i \big( \tau_k^{-i}, \hat{\pi}^i \big) := \mathbb{E}_{\substack{s \sim \rho_{\Pi_{k,i,j}} \\ a^i \sim \hat{\pi}^i(\cdot|s)}} \Big[ A_{\Pi_{k,i,j}}^i(s, a^i) \Big].$$

$L$ aggregates the local signal $A^i$ into a policy-level surrogate by averaging, which represents local surrogate improvement. It satisfies the baseline-zero property $L_{\Pi_{k,i,j}}^i \big( \tau_k^{-i}, \pi_{k,j}^i \big) = 0$, because $\mathbb{E}_{a^i \sim \pi_{k,j}^i(\cdot|s)} [A_{\Pi_{k,i,j}}^i(s, a^i)] = 0$ as stated in (Zhong et al., 2024).

**Step objective.** $F_{k,i,j}$ scores a candidate update as "local surrogate improvement $L$ minus a KL safety penalty", defined as

$$F_{k,i,j}(\hat{\pi}^i) := L_{\Pi_{k,i,j}}^i \big( \tau_k^{-i}, \hat{\pi}^i \big) - C_{k,i,j} \, D_{\mathrm{KL}}^{\max}\big( \pi_{k,j}^i, \hat{\pi}^i \big),$$

where $\varepsilon_{k,i,j} := \max_{s,a} |A_{\Pi_{k,i,j}}(s, a)|$, $C_{k,i,j} := \frac{4\gamma \, \varepsilon_{k,i,j}}{(1-\gamma)^2}$, and $D_{\mathrm{KL}}^{\max}(\mu, \nu) := \sup_s \mathrm{KL}\big( \mu(\cdot \mid s) \,\|\, \nu(\cdot \mid s) \big)$. Taking the current iterate as the candidate reveals that it possesses the baseline-zero property: $F_{k,i,j}(\pi_{k,j}^i) = 0$. This is because $L$ equals zero due to the zero-mean property of the single-agent advantage under the policy $\pi_{k,j}^i$, and $D_{\mathrm{KL}}^{\max}(\mu, \mu) = 0$.

## C.2   THEORETICAL PROOFS

**Proposition 1.** In round $k$, when agent $\pi_{k,j}^i$ updates to $\pi_{k,j+1}^i$, the new joint policy is $\Pi_{k,i,j+1} := (\tau_k^{-i}, \pi_{k,j+1}^i)$, and the performance satisfies

$$J(\Pi_{k,i,j+1}) \ \geq \ J(\Pi_{k,i,j}) \ + \ L_{\Pi_{k,i,j}}^i(\tau_k^{-i}, \pi_{k,j+1}^i) \ - \ C_{k,i,j} \, D_{\mathrm{KL}}^{\max}(\pi_{k,j}^i, \pi_{k,j+1}^i). \tag{3}$$

**Proof of Proposition 1.**   Set old $= \Pi_{k,i,j}$ and new $= \Pi_{k,i,j+1} = (\tau_k^{-i}, \pi_{k,j+1}^i)$. Apply Lemma 6 in (Zhong et al., 2024) with $\pi = $ old and $\bar{\pi} = $ new. Because only agent $i$ changes, the multi-agent surrogate in Lemma 6 equals to $L_{\Pi_{k,i,j}}^i(\tau_k^{-i}, \pi_{k,j+1}^i)$, and the max-conditional KL becomes $D_{\mathrm{KL}}^{\max}(\pi_{k,j}^i, \pi_{k,j+1}^i)$. With $C_{k,i,j} = \frac{4\gamma \, \varepsilon_{k,i,j}}{(1-\gamma)^2}$ and $\varepsilon_{k,i,j} = \max_{s,a} |A_{\Pi_{k,i,j}}(s,a)|$, this yields exactly inequality 3. $\qquad\square$

Our analysis leverages part of the theoretical result in Zhong et al. (2024), but instantiates it at a different level of alternation. Lemma 6 in that work bounds the performance difference between an old joint policy $\pi$ and a new joint policy $\bar{\pi}$ by the sum of per-agent surrogate improvements minus KL penalties. In (Zhong et al., 2024), all agents are updated sequentially within a single step of the algorithm, so Lemma 6 is directly applied at this step-level alternation, where one sweep updates all agents once.

Conversely, Alg. 1 utilizes a round-level interleaving approach: during round $k$, all other agents $\tau_k^{-i}$ remain fixed while a single agent $i$ undergoes multiple inner updates, $j = 0, \ldots, K_i - 1$ before switching to the next agent. As a result, at each step, when agent $\pi_{k,j}^i$ transitions to $\pi_{k,j+1}^i$, the updated joint policy becomes $\Pi_{k,i,j+1} := (\tau_k^{-i}, \pi_{k,j+1}^i)$, differing from the previous joint policy solely in the context of agent $i$. Therefore, we proceed incrementally: first, we instantiate Lemma 6 at each inner update of a single agent, which isolates agent $i$'s contribution and derives the surrogate bound inequality in Proposition 1. Second, we sum these inequalities over all inner updates and agent switches. Employing this telescoping argument results in our round-level monotonic improvement guarantee as stated in Theorem 1.

**Theorem 1.** Every step updates in Alg. 1 satisfies $F_{k,i,j}(\pi_{k,j+1}^i) \geq 0$, then for all outer rounds $k$ we have $J(\pi_{k+1}) \geq J(\pi_k)$.

**Proof of Theorem 1.**   Leveraging the baseline-zero characteristic, $F_{k,i,j}(\pi_{k,j}^i) = 0$, it follows that $\arg\max_{\pi^i} F_{k,i,j}(\pi^i) \geq 0$. Therefore, by Proposition 1 and the update rule,

$$J(\Pi_{k,i,j+1}) - J(\Pi_{k,i,j}) \ \geq \ F_{k,i,j}(\pi_{k,j+1}^i) \ \geq \ 0$$

for every step $(k,i,j)$. Summing these inequalities in order over all steps within round $k$ gives

$$J(\pi_{k+1}) - J(\pi_k) = \sum_{i=1}^n \sum_{j=0}^{K_i-1} \left[ J(\Pi_{k,i,j+1}) - J(\Pi_{k,i,j}) \right] \ \geq \ 0,$$

hence $J(\pi_{k+1}) \geq J(\pi_k)$ for all $k$. $\qquad\square$

**Corollary 1.** The sequence $\{J(\pi_k)\}$ has a limit, denoted as $\bar{J}$, and the set comprised of limit points of $\{\pi_k\}$ is not empty. Additionally, for any convergent subsequence $\{\pi_{k_j}\}_{j \geq 0} : \pi_{k_j} \to \bar{\pi}$, $J(\bar{\pi}) = \bar{J}$.

**Proof of Corollary 1.**   By Theorem 1, the performance sequence $\{J(\pi_k)\}_{k \geq 0}$ is nondecreasing. With discount $\gamma \in [0, 1)$ and bounded rewards $|r| \leq R_{\max}$, every policy satisfies $|J(\pi)| \leq R_{\max}/(1 - \gamma)$, so $\{J(\pi_k)\}$ is bounded above and converges to some $\bar{J} \in \mathbb{R}$. Furthermore, as in (Zhong et al., 2024), the sequence of policies is bounded, hence it admits a convergent subsequence by the Bolzano-Weierstrass Theorem. Therefore, the set of limit points of $\{\pi_k\}$ is nonempty. Let $(\pi_{k_j})_{j \geq 0}$ be any subsequence converging to a limit policy $\bar{\pi}$. By continuity of $J$ in $\pi$, we have

$$J(\bar{\pi}) \ = \ J\left(\lim_{j \to \infty} \pi_{k_j}\right) \ = \ \lim_{j \to \infty} J(\pi_{k_j}) \ = \ \bar{J}.$$

$\qquad\square$

# D    DETAILS IN METHOD

## D.1    DESCRIPTION OF ALGORITHM 3

In Alg. 3, $RL(\theta \mid \mathcal{J}, \cdot, \cdot, \mu)$ denotes GRPO-type single agent RL based on the objective $\mathcal{J}$ defined in equation 2 and hyperparameter $\mu$, where the first argument $\theta$ is the parameter to be updated.

- When updating the *navigator*, the interactor parameters are frozen. In this case, the input $\theta_n$ serves as the initial setting for the navigator, whereas $\theta_{\text{frozen}}$ pertains to the fixed interactor:

$$RL(\theta \mid \mathcal{J}, \theta_n, \theta_{\text{frozen}}, \mu).$$

- When updating the *interactor*, the navigator parameters are frozen. In this case, the input $\theta_i$ serves as the initial setting for the interactor, whereas $\theta_{\text{frozen}}$ pertains to the fixed navigator:

$$RL(\theta \mid \mathcal{J}, \theta_{\text{frozen}}, \theta_i, \mu).$$

Below, we take the case of freezing the interactor and updating the navigator as an example.

---

**Algorithm 3:** GRPO-Type policy optimization $RL(\theta \mid \mathcal{J}, \theta_n, \theta_{\text{frozen}}, \mu)$

---

```
// Here we take the case of freezing the interactor and
   updating the navigator as an example.
```
**Input:** Dataset $\mathcal{D}$, current policy $\theta_n$, frozen model $\theta_{\text{frozen}}$, hyperparameters $\mu = (M, B, K)$;
Initialize $\theta \leftarrow \theta_n$;
**for** *iteration* $= 1 \ldots M$ **do**
    Initialize reference policy $\theta_{\text{ref}} \leftarrow \theta$;
    **for** *step* $= 1 \ldots B$ **do**
        Sample a batch $\mathcal{D}_b$ from $\mathcal{D}$;
        Update the old policy $\theta_{\text{old}} \leftarrow \theta$;
        Sample $K$ outputs $\{r_k\}_{k=1}^{K} \sim \pi_{\theta_{\text{old}}, \theta_{\text{frozen}}}(\cdot \mid I, H_t, X_t)$ for each question in the batch;
        Compute rewards $\{R_k\}_{k=1}^{K}$ via reward model;
        Compute GRPO-type advantages: $A_k \leftarrow \frac{R_k - \mu}{\sigma}$ over the group;
        **Online reweighting:** discard samples with low quality and resample to refill the batch;
        Update the policy model by maximizing the objective: $\theta \leftarrow \arg\max_\theta \mathcal{J}(\theta, \theta_{\text{frozen}})$;
**return** *Updated parameters* $\theta$;

---

## D.2    DETAILS OF ONLINE REWEIGHTING

After computing per-sample advantages $A_k$ on the unfiltered group, we apply an online reweighting step: (1) filter the raw batch with a rule $R(\cdot)$ to keep high-quality samples; (2) randomly resample with replacement from the retained set until the batch reaches the predefined target size $b$, as detailed in Alg. 4.

Different from simply discarding low-quality samples, our online reweighting yields a more stable batch size and gradient scale, does not alter the within-group statistics used to normalize advantages (thus remaining unbiased), and maintains sufficient training signal when few samples are retained, so the effective batch does not shrink. From a reweighting perspective, this is equivalent to assigning

zero weight to discarded samples and stochastic weights to the retained ones, amplifying high-quality samples with minimal implementation complexity.

---

**Algorithm 4:** Online Reweighting for SWIRL

---

**Input:** Batch $\mathcal{B} = \{r_j\}_{j=1}^{J}$, selection rule $\mathcal{R}(\cdot)$, target batch size $b$;

Initialize $\mathcal{S} \leftarrow \emptyset$;

// Rule-based filtering: keep only high-quality samples.

**for** $j = 1 \ldots J$ **do**

    **if** $\mathcal{R}(r_j) = retain$ **then**

        $\mathcal{S} \leftarrow \mathcal{S} \cup \{r_j\}$;

$\widehat{\mathcal{B}} \leftarrow \mathcal{S}$;

// Resample with replacement until reaching target size $b$.

**while** $|\widehat{\mathcal{B}}| < b$ **do**

    Sample $y \sim \text{Sampler}(\mathcal{S})$;

    $\widehat{\mathcal{B}} \leftarrow \widehat{\mathcal{B}} \cup \{y\}$;

**return** $\widehat{\mathcal{B}}$;

---

### D.3 BENEFITS OF SWIRL

Different from MARL methods such as HAPPO (Zhong et al., 2024) (Fig. 2a), A2PO (Wang et al., 2023), and MARFT (Liao et al., 2025) (Fig. 2b), our approach (Fig. 2c) offers four key advantages:

1. **Seamless compatibility.** SWIRL reformulates complex MARL tasks into an alternating sequence of simpler single-agent reinforcement learning problems. This design naturally integrates with modern distributed single-agent RL frameworks (e.g., VeRL (Sheng et al., 2025), OPEN-RLHF (Hu et al., 2024)), eliminating intrusive modifications to communication protocols and enabling rapid adaptation to future efficient RL frameworks. Consequently, our divide-and-conquer alternating solution fundamentally resolves the integration challenges between MARL and contemporary LVLM-based training pipelines.

2. **Resource-friendly scalability.** Consider a system with $N$ agents, each with model size $|\theta_i|$. Table 1 quantitatively summarizes the number of actor parameters loaded on the device during training. In practice, A2PO, HAPPO, and MARFT keep all agents resident on the training device for local rollout. Under this setting, these methods typically load all actor parameters in the training device during learning, leading to a total parameter size of $\sum_{i=1}^{N} |\theta_i|$ and memory consumption that scales linearly with the number of agents ($O(N)$). In contrast, SWIRL only loads the currently updated agent model locally, while executing the remaining agents as "Model-as-a-Service" modules remotely. As a result, the memory usage of the training device remains constant regardless of $N$ (i.e., $O(1)$), greatly reducing hardware requirements and improving scalability for large-scale multi-agent systems.

3. **Adaptation to heterogeneity.** Our method allows agents to adopt diverse model architectures and training configurations, such as different numbers of training steps or heterogeneous datasets. Because SWIRL decomposes multi-agent training into per-agent optimization, it enables online, agent-conditioned reweighting that adapts to each agent's distribution by filtering samples that are low-quality for that agent. As a result, heterogeneous agents receive tailored, high-confidence training signals; experiments on Appendix F.4 empirically confirm reduced uninformative or misleading samples and improved overall training performance.

4. **Stability.** Fig. 4 shows consistent performance improvement across training phases. This stability arises from the overall SWIRL training mechanism, in particular from the combination of round-level alternating updates and online reweighting. Appendices F.1 and F.4 show that coordination improves over rounds, while Appendix F.3 demonstrates that allocating the same budget to more alternating rounds yields better results than deeper single-agent updates. While SWIRL does not aim to eliminate non-stationarity in fully concurrent MARL, it provides stable and theoretically grounded optimization in our sequential multi-agent framework.

# E    EXPERIMENT DETAILS

## E.1    TRAINING DETAILS

**Training Data Collection.** We construct our two-stage training dataset based on the AITW (Rawles et al., 2023) (using the expanded version from AITZ (Zhang et al., 2024)) and AMEX (Chai et al., 2024) (adopting the Aguvis variant (Xu et al., 2024)), as both provide richer low-level annotations. In Stage 1, we leverage Qwen2.5-VL-3B to generate 8 rollouts for each low-level instruction, then select 1,500 samples whose average reward falls within the range $[0.3, 0.4]$. These high-quality samples are used in two ways: (1) the low-level instruction data is employed as RL training data for the Interactor, and (2) the reasoning traces associated with each sample are used to construct Chain-of-Thought supervised fine-tuning data with high-level instructions for the Navigator. In Stage 2, we utilize the multi-agent system trained in Stage 1 to generate 8 rollouts for each high-level instruction, filtering for 2,000 samples with an average reward below 0.6 and variance more than 0.175. Notably, samples in Stage 2 include only high-level instructions and do not contain additional semantic annotations. In total, we curate 3,500 high-quality training samples across both stages, supporting robust chain-of-thought reasoning and effective RL optimization for both agents.

**Action Space.** Following the action space design style of UI-Tars (Qin et al., 2025), we define dataset-specific action spaces for different GUI benchmarks, as shown in Table 8.

**Prompt.** Fig. 8 illustrates the prompts used for the Navigator and Interactor.

**Wall Time.** During the interleaved training stage, each round takes 255 minutes of wall-clock time: training the Navigator model requires 140 minutes, training the Interactor model requires 85 minutes, and 35 minutes are spent on data preparation, agent deployment, environment initialization, and other overhead. The total wall-clock time for our 20-round setup is 85 hours.

## E.2    EVALUATION DETAILS

**Metrics.** Following prior work (Wu et al., 2024b; Luo et al., 2025), we evaluate our models using three standard metrics for GUI agents: Type, GR, and SR, which assess the accuracy of action type prediction, coordinate prediction, and step success rate, respectively. Type measures the exact match between the predicted action type (e.g., click, scroll) and the ground truth. GR evaluates the performance of GUI grounding in downstream tasks. SR (step-wise success rate) is computed by considering a step correct only if both the predicted action type and all associated arguments (e.g., coordinates for a click action) match the ground truth. For click-based actions (e.g., click, long_press), the model must predict both the action type and the target coordinates (x, y). When the ground-truth bounding box is available, a prediction is considered correct if its coordinates fall within the box. If no bounding box is provided, or the prediction lies outside it, correctness is determined by whether the predicted coordinates are within $14\%$ of the screen width from the ground-truth position. For type-based actions (e.g., type, open_app), both the action type and content must match the ground truth. We compute the F1 score between the predicted and reference text, and an action is considered correct if F1 $> 0.5$. For scroll actions, the predicted direction argument (up, down, left, or right) must exactly match the ground truth. For all other actions (e.g., press_enter), correctness requires an exact match between the predicted action and the ground truth.

**Baselines.** To ensure a fair comparison of cross-domain generalization, we select baseline models that have not been trained on the corresponding training sets of the evaluation domains. For high-level tasks, the comparison includes four categories of models: the proprietary GPT-4o (OpenAI, 2024); the state-of-the-art OS-Atlas-4B/7B (Wu et al., 2024b), trained via supervised fine-tuning (SFT) on large-scale GUI grounding datasets; models trained with reinforcement fine-tuning (RFT) on GUI grounding datasets, including UI-R1-3B, UI-R1-E-3B (Lu et al., 2025), GUI-R1-3B/7B (Luo et al., 2025), and ReGUIDE-7B (Lee et al., 2025); and a multi-agent approach employing GPT-5 (OpenAI, 2025) as the planner to translate high-level instructions into low-level actions. For low-level tasks, the baselines consist of GPT-4o, OS-Atlas-4B/7B, and RFT-trained

GUI grounding models, including UI-R1-3B, UI-R1-E-3B, GUI-R1-3B/7B, ReGUIDE-7B, and SE-GUI-7B (Yuan et al., 2025).

**Evaluation Datasets.** **AndroidControl** (Li et al., 2024b) is a mobile control dataset in which each GUI interaction trajectory is annotated with both coarse-grained high-level instructions and fine-grained low-level instructions, along with detailed XML metadata from which the bounding boxes of individual UI elements can be parsed. For click actions, we iterate over all bounding boxes in the XML, identify those containing the ground-truth coordinates, and select the smallest one by area as the candidate bounding box for click action evaluation. **GUIOdyssey** (Lu et al., 2024) consists of tasks involving cross-application operations, posing a significant challenge for models' planning abilities. In its latest release[3], each click action is supplemented with the bounding box of the corresponding UI element obtained via SAM2 (Ravi et al., 2024) segmentation, which we also use for click action evaluation. Notably, the latest version contains $8,834$ samples, compared to $7,735$ in the earlier release[4], with changes in the test set size. To ensure consistency with other baselines, we use the Test-Random split from the earlier version. **GUI-Act-Web** (Chen et al., 2024) contains web-based interaction data. To better assess GUI manipulation capabilities, we remove several QA-style samples from the test set and retain the remaining samples for evaluation. Bounding boxes are not used for click action evaluation in this dataset. **OmniAct** (Kapoor et al., 2024) includes both web and desktop interaction data. Due to action space compatibility constraints, we filter out samples whose original action space involves hotkeys and keep the remaining samples for evaluation. Similarly, bounding boxes are not used for click action evaluation in this dataset.

### E.3 MATHEMATICS DOMAINS

**Implementation Details.** We design a dual-agent framework in which one agent acts as the *Teacher*, providing a concise outline of the problem-solving approach, and the other acts as the *Student*, generating the final solution by incorporating the Teacher's guidance. For the 3B implementation, consistent with MARFT (Liao et al., 2025), both agents are initialized from Qwen2.5-Coder-3B-Instruct (Hui et al., 2024). For the 7B implementation, following ReMA (Wan et al., 2025), both agents are initialized from Qwen2.5-7B-Instruct (Team, 2024). The detailed prompt formulations for both agents are provided in Fig. 9. For the reward function, we assign a reward of 1 if the generated answer is correct and 0 otherwise, i.e., $R_x = \begin{cases} 1, & \text{if answer is correct,} \\ 0, & \text{otherwise.} \end{cases}$ . For online reweighting, we define the rule $\mathcal{R}$ as follows: we retain samples whose average reward across multiple rollouts lies strictly between 0.2 and 0.8, i.e., $\mathcal{R} = \begin{cases} \text{retain,} & 0.2 < \overline{R_x} < 0.8, \\ \text{discard,} & \text{otherwise.} \end{cases}$ , where $\overline{R_x}$ denotes the mean reward of sample $x$ over all rollouts.

**Training.** We train the dual-agent framework on the MATH (Hendrycks et al., 2021) training set, which contains $7,500$ samples, by directly initiating Stage 2 of the SWIRL alternating-update procedure and bypassing the warm-up stage. In each round, the Teacher is updated first, followed by the Student, with each agent trained for 1 epoch per round over a total of 10 rounds. Both agents use a batch size of 128 and a learning rate of $1 \times 10^{-6}$ . The Teacher and Student perform 4 and 8 rollouts per sample, respectively. Training is conducted on $16\times$ NVIDIA A800 GPUs, with half allocated to deploying the Student as a vLLM-based (Kwon et al., 2023) inference service and the remaining half for model training.

**Evaluation.** For the 3B setting, following MARFT (Liao et al., 2025), we evaluate in-domain, cross-domain, and out-of-domain reasoning on MATH500 (Hendrycks et al., 2021), CMATH (Wei et al., 2023), and GSM8K (Cobbe et al., 2021), respectively. For the 7B setting, consistent with ReMA (Wan et al., 2025), we evaluate in-domain performance on MATH500 (Hendrycks et al., 2021) and out-of-domain generalization on GSM8K (Cobbe et al., 2021), Minerva-Math (Lewkowycz et al., 2022), OlympiadBench (He et al., 2024), AIME24[5], AMC23[6], and

---

[3] https://huggingface.co/datasets/hflqf88888/GUIOdyssey
[4] https://huggingface.co/datasets/OpenGVLab/GUIOdyssey
[5] https://huggingface.co/datasets/AI-MO/aimo-validation-aime
[6] https://huggingface.co/datasets/AI-MO/aimo-validation-amc

Table 8: Action spaces used for each dataset.

| Dataset | Action Space |
|---|---|
| AITW, AMEX | ```click(point='(x1, y1)')```
```type(content='xxx')```
```scroll(direction='down\|up\|right\|left')```
```press_home()```
```press_back()```
```press_enter()```
```finished()``` |
| AndroidControl | ```click(point='(x1, y1)')```
```long_press(point='(x1, y1)')```
```type(content='xxx')```
```scroll(direction='down\|up\|right\|left')```
```open_app(app_name='xxx')```
```press_home()```
```press_back()```
```wait()```
```finished()``` |
| GUIOdyssey | ```click(point='(x1, y1)')```
```long_press(point='(x1, y1)')```
```type(content='xxx')```
```scroll(direction='down\|up\|right\|left')```
```press_home()```
```press_back()```
```press_appselect()```
```error(content='xxx')```
```finished()``` |
| GUI-Act-Web | ```click(point='(x1, y1)')```
```scroll(direction='down\|up')``` |
| OmniAct-Web) | ```click(point='(x1, y1)')```
```rightclick(point='(x1, y1)')```
```scroll(direction='down\|up')``` |
| OmniAct-Desktop | ```click(point='(x1, y1)')```
```rightclick(point='(x1, y1)')```
```doubleclick(point='(x1, y1)')```
```moveto(point='(x1, y1)')```
```scroll(direction='down\|up')``` |

Gaokao2023en (Zhang et al., 2023). All benchmarks are evaluated using *pass@1 accuracy*, where the model produces a single solution attempt that is directly judged against the ground truth.

# F ABLATION STUDY

## F.1 CO-EVOLUTION OF NAVIGATOR AND INTERACTOR

We investigate the impact of the proposed interleaved update on individual agents within the multi-agent framework. For the Interactor, Table 9 compares its performance on low-level tasks with and without Stage 2 interleaved updates. The results show a substantial improvement of 4.89 points with Stage 2, particularly in SR (a metric that directly measures the correctness of individual actions and thus serves as a more critical indicator of the Interactor's precision), which increases by nearly 7 points. For the Navigator, direct evaluation is less straightforward because its outputs are low-level instructions rather than executable actions. To address this, we adopt an indirect evaluation approach: we feed the Navigator's outputs into a fully trained Interactor and assess the resulting task performance. As shown in Table 10, pairing the Interactor with a Navigator trained using

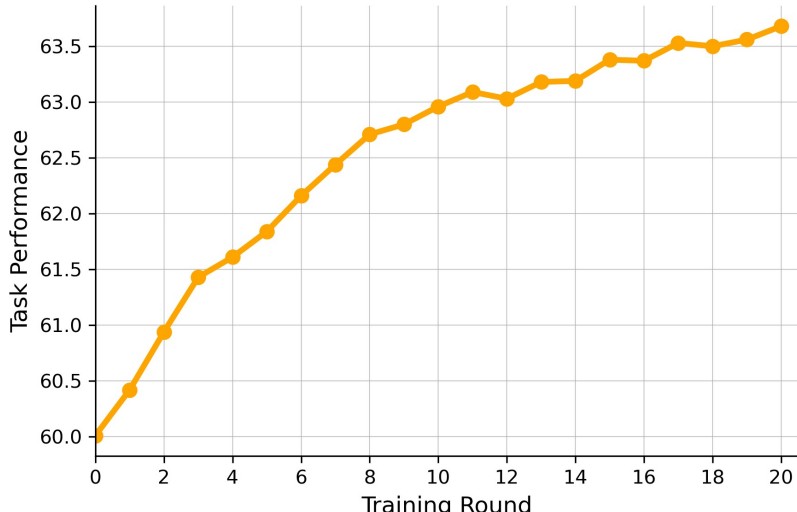

Figure 4: SWIRL training dynamics showing steady performance gains and stability across rounds. Task performance is measured as the average score across all high-level benchmarks, including AndroidControl-High and GUIOdyssey.

Stage 2 interleaved updates delivers consistently higher performance on high-level tasks, raising the overall score from 56.32 to 63.68 and yielding notable improvements in both GR and SR across benchmarks. This demonstrates that the updated Navigator produces more detailed and accurate low-level instructions from high-level goals. Overall, these findings provide strong evidence that interleaved updates effectively enhance each agent's ability to fulfill its specific role, enabling them to co-evolve and achieve better collaborative performance within the multi-agent system.

Table 9: Effect of interleaved updates on Interactor performance in low-level tasks.

| Training Strategy | GUI-Act-Web | | | OmniAct-Web | | | OmniAct-Desktop | | | AndroidControl-Low | | | Overall |
|---|---|---|---|---|---|---|---|---|---|---|---|---|---|
| | Type | GR | SR | Type | GR | SR | Type | GR | SR | Type | GR | SR | |
| Stage 1 | 94.31 | **88.22** | 77.00 | 89.22 | 81.62 | 72.40 | 91.38 | 74.45 | 68.03 | **85.48** | 71.91 | 68.87 | 80.24 |
| Stage 1 + Stage 2 | **95.00** | 87.85 | **84.85** | **94.52** | **81.67** | **77.32** | **94.65** | **77.09** | **72.97** | 84.62 | **92.20** | **78.81** | **85.13** |

Table 10: Performance of multi-Agent systems on high-level tasks with Navigators trained with and without interleaved updates.

| Stage 2 Training | AndroidControl-High | | | GUIOdyssey | | | Overall |
|---|---|---|---|---|---|---|---|
| | Type | GR | SR | Type | GR | SR | |
| ✗ | 63.16 | 54.65 | 39.91 | 73.19 | 60.49 | 46.51 | 56.32 |
| ✓ | **66.72** | **71.19** | **51.24** | **74.87** | **66.39** | **51.65** | **63.68** |

## F.2 STABILITY AND POTENTIAL

As illustrated in Fig. 4, even with only $2,000$ samples used during the SWIRL's stage 2, the proposed training paradigm consistently enhances the model's generalization capability. The model's performance improves steadily with each training round, demonstrating the stability of the alternating optimization process. Furthermore, its out-of-domain generalization continues to increase in the later rounds, suggesting that the performance upper bound has not yet been reached. These results highlight the effectiveness and scalability of SWIRL for robust multi-agent training in GUI navigation tasks.

### F.3 SYNERGY AND INDIVIDUAL COMPETENCE IN MULTI-AGENT TRAINING

In multi-agent training, overall system performance depends not only on enhancing the capabilities of individual agents but also on achieving effective coordination among them. To disentangle and quantify the relative contributions of these two factors, we vary the number of alternating training rounds (i.e., the frequency and intensity of inter-agent coordination) and the number of epochs per round (i.e., the depth of single-agent training), while keeping the total number of training epochs constant. As shown in Fig. 5(a) and Table 11, increasing the number of alternating rounds, thereby providing more opportunities for inter-agent interaction and iterative policy refinement, consistently leads to greater performance gains than allocating the same budget to deeper single-agent training within fewer rounds. At the same time, Fig. 5(b) indicates that increasing the training depth per round still brings incremental benefits, demonstrating that single-agent optimization remains valuable. Taken together, these results suggest that, under a fixed training budget, inter-agent coordination is the primary driver of performance improvement, while individual agent refinement plays a complementary yet meaningful role.

Table 11: Effect of the rounds-epochs schedule under a fixed training budget. Here, 'Epochs/round' denotes the number of epochs trained within each round. The total training epochs are held constant across settings.

| Rounds | Epochs/round | AndroidControl | | | GUIOdyssey | | | Overall |
|---|---|---|---|---|---|---|---|---|
| | | Type | GR | SR | Type | GR | SR | |
| 2 | 10 | 65.74 | 70.42 | 49.88 | 73.13 | 64.85 | 49.27 | 62.21 |
| 5 | 4 | 65.26 | **70.94** | 49.81 | 73.85 | 65.21 | 49.98 | 62.51 |
| 10 | 2 | **66.00** | 70.91 | **50.56** | **74.03** | **65.71** | **50.58** | **62.96** |

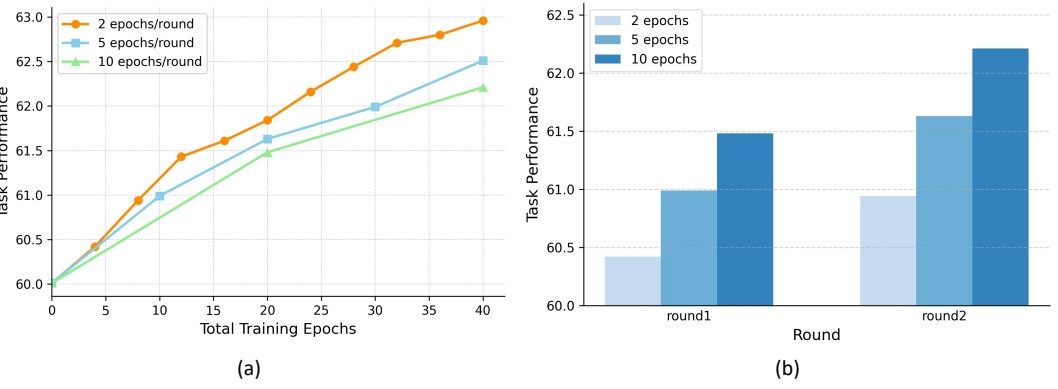

(a)                                                                 (b)

Figure 5: Effect of the training schedule on SWIRL performance with varying numbers of rounds and epochs per round. (a) Training curves under a fixed total number of training epochs, comparing different numbers of epochs per round. (b) Performance comparison after each round for different epochs-per-round settings.

### F.4 EFFECTIVENESS ANALYSIS OF ONLINE REWEIGHTING

To assess the impact of the online reweighting mechanism, we conduct comparative experiments with and without weighted resampling. As shown in Fig. 7(a), models trained without online reweighting perform significantly worse and even exhibit a degradation trend, underscoring the necessity of this mechanism. We hypothesize that online reweighting dynamically prioritizes informative samples, ensuring that they exert greater influence during training, which is essential for the stable improvement of multi-agent systems. Further analysis, as illustrated in Fig. 6, reveals two important findings. First, the Interactor filters out approximately four times as many samples as the Navigator ($\sim$75 vs. $\sim$18), indicating a substantial difference in how the same dataset contributes to the learning process of each agent. The online reweighting strategy thus adapts to the evolving capabilities of each agent, assigning dynamic weights to those training samples most beneficial at

each stage. Second, it is important to recognize that not all samples filtered for being completely incorrect are attributable solely to the deficiencies of a single agent. Some errors may arise from noisy data or from mistakes originating with another agent (e.g., if the Planner generates an erroneous instruction, the Executor may repeatedly fail to execute the correct action). In these cases, the online reweighting mechanism helps to exclude uninformative or misleading samples, thereby further improving the robustness and effectiveness of the training process.

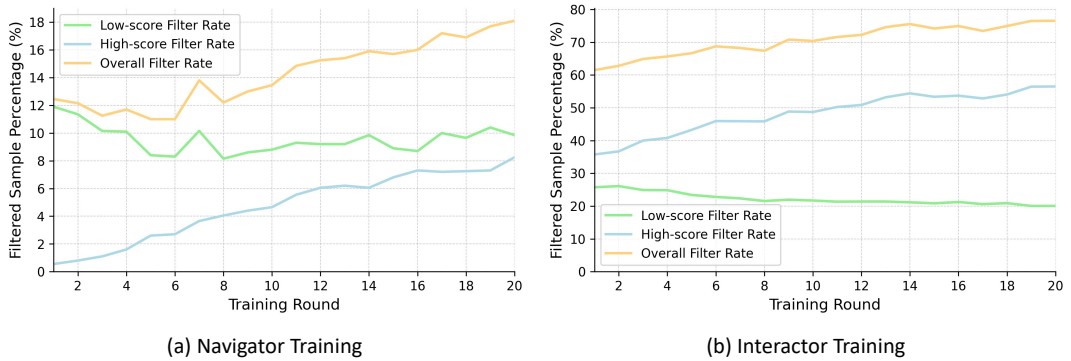

(a) Navigator Training           (b) Interactor Training

Figure 6: Filtered sample rates with online reweighting in SWIRL.

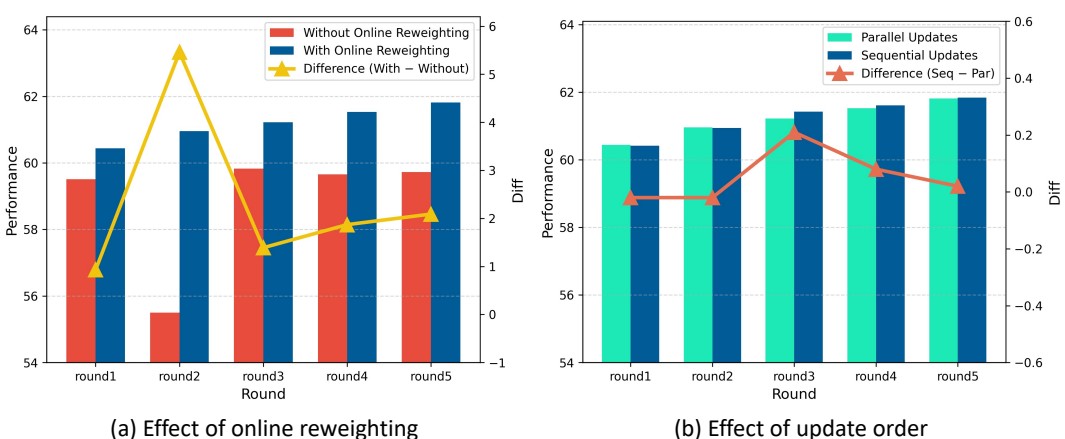

(a) Effect of online reweighting           (b) Effect of update order

Figure 7: Ablation results on SWIRL zero-shot performance for AndroidControl-High and GUIOdyssey. (a) Online reweighting vs. no reweighting. (b) Sequential vs. parallel updates. Performance differences are indicated by lines.

## F.5 SEQUENTIAL UPDATES VERSUS PARALLEL UPDATES

Our default SWIRL implementation employs a strictly sequential update scheme, in which the Navigator is updated first in each training round, followed by the Interactor. We also explore a parallel update strategy, where both agents are updated simultaneously within each round, rather than following a fixed order, i.e., $\begin{cases} \theta_n^{(k+1)} \leftarrow RL(\theta_n | \mathcal{J}, \theta_n^{(k)}, \theta_i^{(k)}, \mu_n) \\ \theta_i^{(k+1)} \leftarrow RL(\theta_i | \mathcal{J}, \theta_n^{(k)}, \theta_i^{(k)}, \mu_i) \end{cases}$. This approach aligns with the update mechanism of Jacobian ADMM (Yang et al., 2022), which relaxes the sequential constraints in standard ADMM and allows for simultaneous updates, i.e., $\begin{cases} x^{k+1} := \arg\min_x \mathcal{L}_\rho(x, y^k, \lambda^k) \\ y^{k+1} := \arg\min_y \mathcal{L}_\rho(x^k, y, \lambda^k) \end{cases}$, thereby increasing training efficiency. As shown in Fig. 7(b), parallel updates can achieve performance comparable to or surpassing that of sequential updates, even without strict alternation. This finding suggests that relaxing the sequential constraint in alternating multi-agent training can improve efficiency without sacrificing model quality.

## F.6 EXTENDING SWIRL TO A THREE-AGENT SYSTEM

To investigate the effectiveness of SWIRL in settings with more than two agents, we extend the original Navigator-Interactor pair by introducing an additional Summarizer agent, forming a three-agent architecture. At each step, the Summarizer converts up to four historical screenshots into a concise textual description summarizing the GUI trajectory, which is then passed as input to the Navigator. The Navigator and Interactor retain the same roles as described in Sec. 2.2: the Navigator produces low-level instructions, and the Interactor executes them as atomic GUI actions. Training this three-agent system follows the SWIRL framework exactly. We alternate RL updates over the three agents in the sequence Summarizer-Navigator-Interactor, freezing the other agents when one is being optimized. Owing to SWIRL's $O(1)$ actor memory footprint, the three-agent configuration uses the same hardware as the two-agent setup and requires no additional devices. Specifically, we employ 8 A800 GPUs to train the active agent and another 8 A800 GPUs to deploy the remaining two agents during each update. As shown in Table 12, SWIRL improves the overall average score from 53.56 to 62.79 (+9.23 points), with consistent gains across all three high-level benchmarks. The results empirically show that SWIRL scales beyond two agents and can support more complex multi-agent architectures in practice.

Table 12: Three-agent performance on in-domain (AITZ) and out-of-domain (AndroidControl, GUI-Odyssey) GUI benchmarks.

| | AITZ | | | AndroidControl | | | GUI-Odyssey | | | Overall |
|---|---|---|---|---|---|---|---|---|---|---|
| | TYPE | GR | SR | TYPE | GR | SR | TYPE | GR | SR | |
| Baseline | 55.25 | 83.01 | 41.98 | 54.80 | 58.59 | 33.12 | 62.53 | 57.49 | 35.23 | 53.56 |
| SWIRL | 60.75 | 89.34 | 53.26 | 61.42 | 70.34 | 45.46 | 71.20 | 65.53 | 47.82 | 62.79 |

## F.7 EFFECTIVENESS OF HIGH-TO-LOW TRANSLATION

To further assess the capability of the Navigator, we evaluate its ability to translate coarse high-level instructions into fine-grained low-level instructions. Since the Navigator produces textual outputs, and these outputs often exhibit legitimate semantic variation (e.g., paraphrases of "open the Chrome app"), simple string-matching metrics alone are insufficient for reliable evaluation. We therefore adopt three complementary metrics: (i) F1 score against the ground-truth low-level instruction; (ii) Human evaluation on 200 sampled cases (100 per dataset), where model-blind annotators assign a 0-1 alignment score; (iii) Execution-based evaluation, where we measure whether Interactor can successfully execute the Navigator's generated instruction. The results show that while the Navigator does not always produce the text most closely aligned with the ground-truth instruction according to human-judged similarity, it consistently yields higher execution success when paired with the Interactor. An illustrative example is provided in Fig. 13 of Appendix H: although the Navigator's generated instruction is not the closest textual match to the ground truth, it preserves the task intent in a form that the Interactor reliably interprets and successfully executes. This suggests that the Navigator does not need to generate perfectly precise low-level instructions for literal execution; rather, effective coordination emerges because the Navigator and Interactor operate with compatible representations and have learned to interpret each other's outputs, enabling robust multi-agent execution.

Table 13: Comparison of the ability to translate high-level instructions into low-level instructions.

| model | AndroidControl | | | GUI-Odyssey | | |
|---|---|---|---|---|---|---|
| | F1 | human-eval | Interactor SR | F1 | human-eval | Interactor SR |
| Qwen2.5-VL-3B | 19.75 | 0.44 | 44.50 | 20.18 | 0.36 | 40.14 |
| Qwen2.5-VL-7B | 20.92 | 0.46 | 47.70 | 22.72 | **0.50** | 43.84 |
| UI-Tars-1.5-7B | 15.80 | 0.52 | 49.30 | 13.96 | **0.50** | 48.56 |
| GPT-5 | 30.71 | **0.56** | 49.53 | 23.79 | 0.35 | 44.21 |
| Navigator | **36.65** | 0.54 | **51.24** | **37.56** | 0.36 | **51.65** |

## F.8 HIGH-LEVEL TASK IN-DOMAIN PERFORMANCE

Our training dataset consists of 3,500 samples collected from AMEX and AITW, with the latter using the curated and extended AITZ version. We evaluate the in-domain capability of our model on the AITZ test set. As shown in Table 14, the Navigator–Interactor system trained with our SWIRL framework achieves a success rate of 62.45, representing the best performance among all compared methods.

Table 14: Performance comparison on AITZ benchmark.

| Model | SR |
| --- | --- |
| AUTO-UI | 34.46 |
| AUTO-UI+CoAT | 47.69 |
| CogAgent | 44.52 |
| CogAgent+CoAT | 53.28 |
| **Navigator + Interactor** | **62.45** |

---

**Navigator**

current_screenshot.png</img>

You are a GUI Planner Agent. Your role is to actively collaborate with the Executor Agent to complete complex GUI navigation tasks. Given a task description, the current screenshot, and the action history from the Executor Agent, your goal is to provide a clear and precise fine-grained instruction for the Executor Agent to help accomplish the task.

## Tools
You need to interact with the Executor Agent by making a tool call:
<tools>
{"type": "function", "function": {"name": "executor_agent", "description": "an Executor Agent capable of executing fine-grained instruction", "parameters": {"type": "object", "properties": {"instruction": {"type": "string", "description": "A clear and precise fine-grained instruction for the executor agent"}}, "required": ["instruction"], "strict": false}}
</tools>

Return a json object with function name and arguments within <tool_call></tool_call> XML tags:
<tool_call>
{"name": <function-name>, "arguments": <args-json-object>}
</tool_call>

## Note
- You should first outline the overall task flow and clarify your next intention. Then, generate a fine-grained, precise, and unambiguous instruction that will guide the Executor Agent to execute one of its available actions: {action_space}.
- Please keep your reasoning within <think> </think> tags, and then output the fine-grained instruction as a tool call in the following format:
<think>...</think><tool_call>...</tool_call>

## User Instruction
{user_high_level_instruction}

**Interactor**

current_screenshot.png</img>

You are a reasoning GUI Executor Agent. Given the attached UI screenshot and the instruction: "{planner_output_instruction}", please determine the next action to fulfill the instruction.

## Action Space
{action_space}

## Note
- Please keep your reasoning in <think> </think> tags brief and focused. Output the final action in <answer> </answer> tags:
<think>...</think><answer>...</answer>

Figure 8: Prompt design for the dual-agent framework in the GUI domain.

| Teacher |
| --- |
| Two LLM agents (Teacher -> Student) collaborate step-by-step to solve math problems. You are the Teacher: Provide only a concise problem-solving strategy without revealing the full solution, and guide the Student to complete the problem.

question: {question} |
| **Student** |
| Two LLM agents (Teacher -> Student) collaborate step-by-step to solve math problems. You are the Student: Follow the original problem and the Teacher's guidance to carry out the necessary operations, and present the final answer within \\boxed{}.

question: {question}

<teacher_response>{teacher_response}</teacher_response> |

Figure 9: Prompt design for the dual-agent framework in the mathematics domain.

# G EXAMPLES

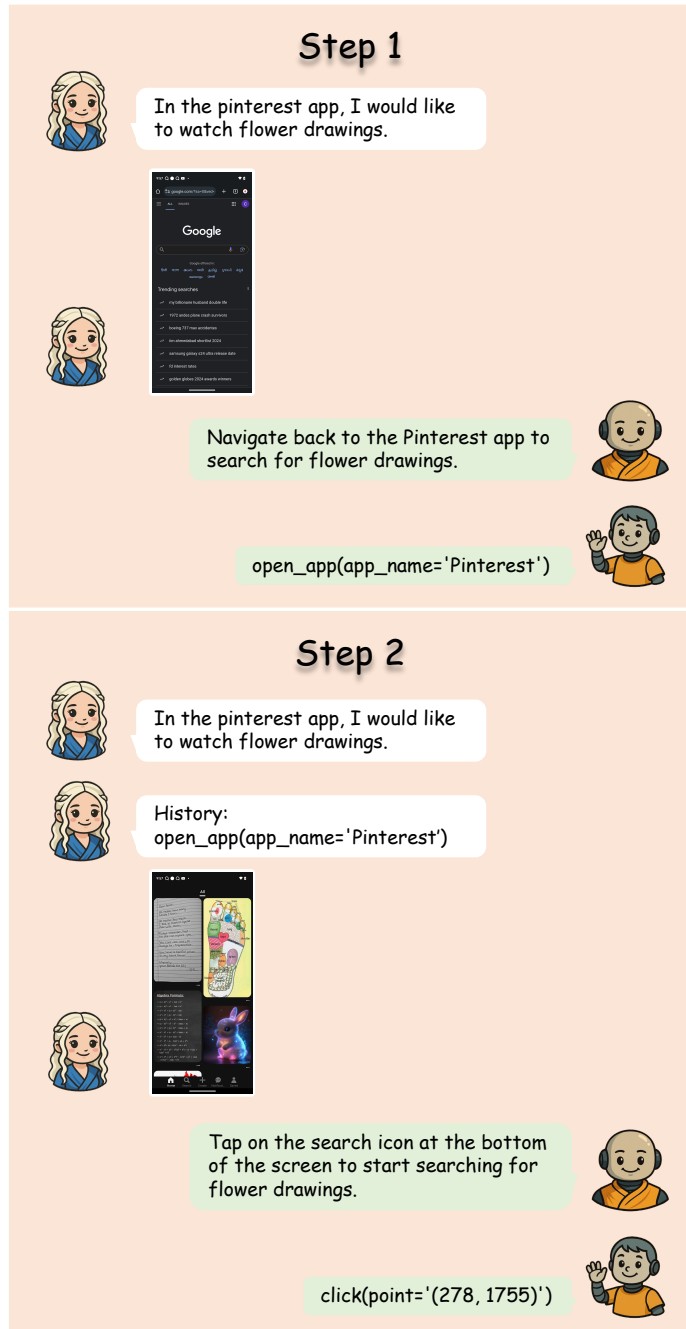

Figure 10: Example of GUI agents collaboratively performing a mobile GUI control task (Part 1 of 3). At each step, the system determines the next action based on the user instruction, the current screenshot, and the action history, iterating until the task is completed.

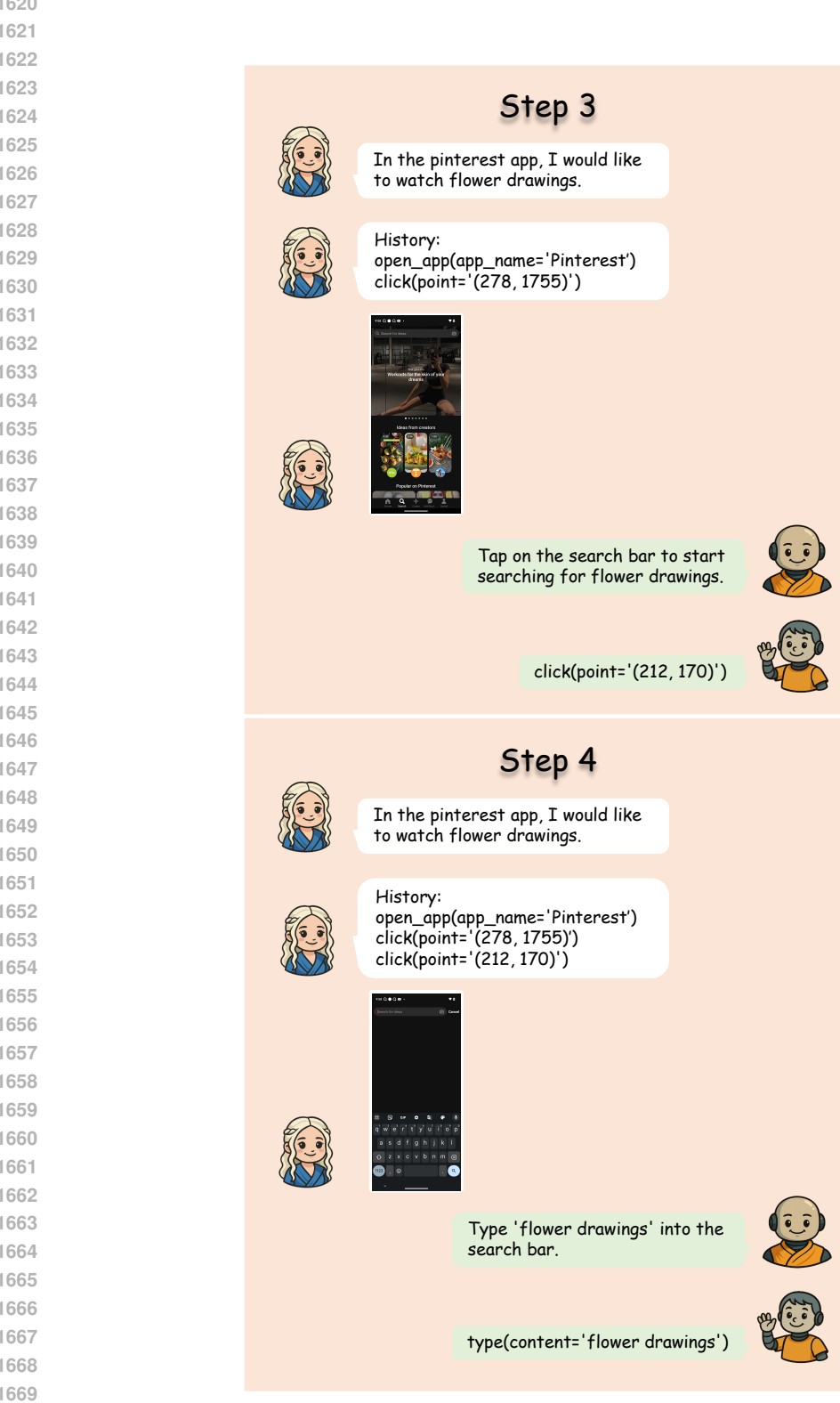

Figure 11: Example of GUI agents collaboratively performing a mobile GUI control task (Part 2 of 3).

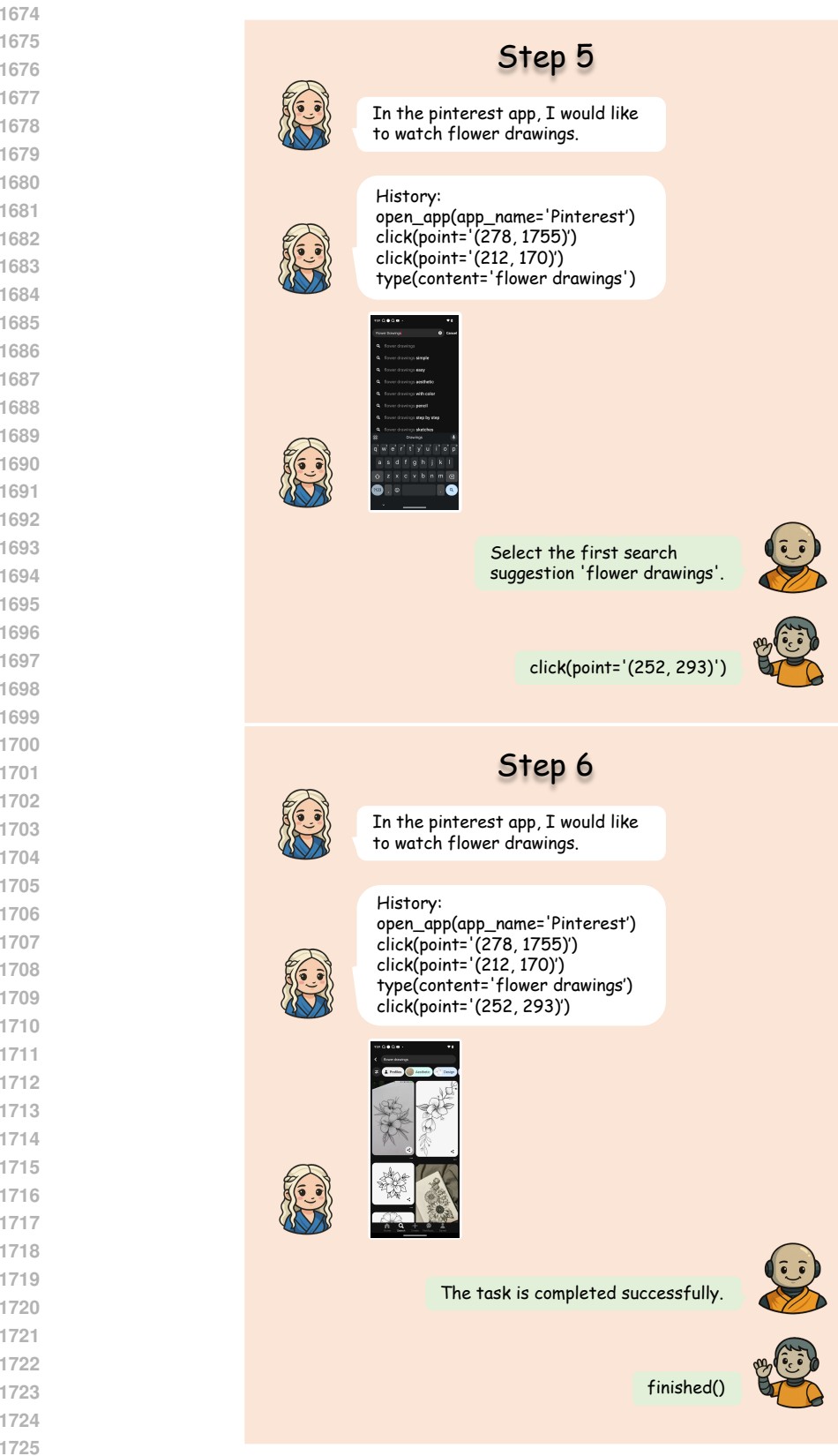

Figure 12: Example of GUI agents collaboratively performing a mobile GUI control task (Part 3 of 3).

# H QUALITITATIVE ANALYSIS

High level instruction: Share the Today Moon Phase information with my friend Natalie Larson at her gmail address, natalie.larson1998@gmail.com.

Low level instruction: Click on the three dots at the top right corner of the screen.
Answer: click(point='(798, 163)')

*Ground-truth label*

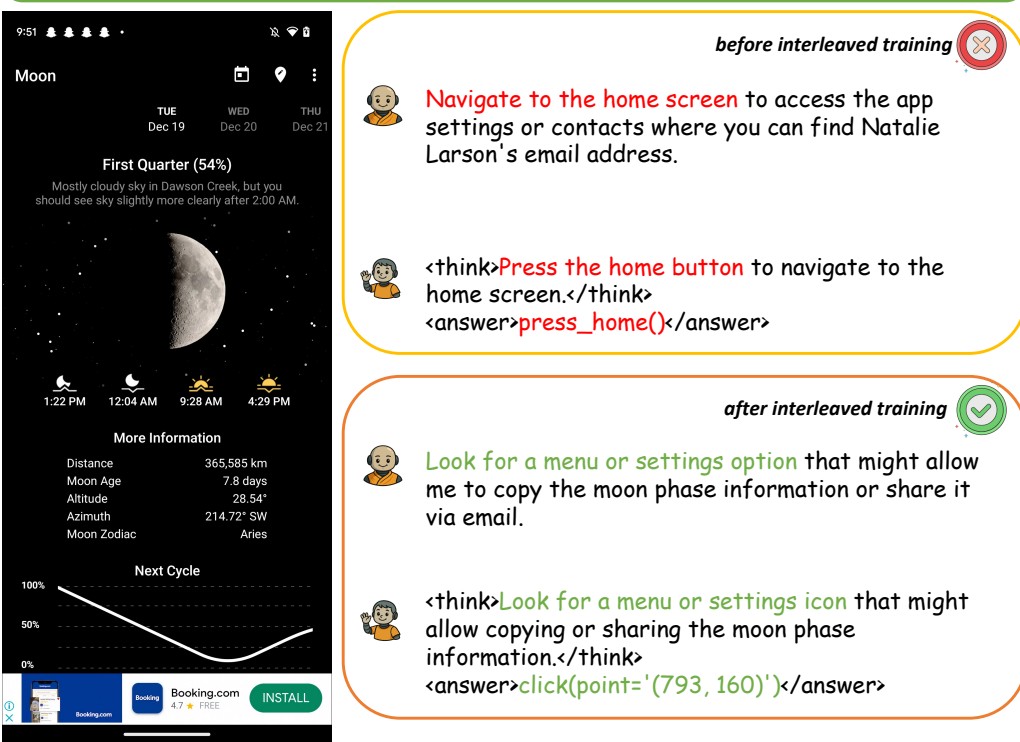

*before interleaved training* ❌

Navigate to the home screen to access the app settings or contacts where you can find Natalie Larson's email address.

<think>Press the home button to navigate to the home screen.</think>
<answer>press_home()</answer>

*after interleaved training* ✅

Look for a menu or settings option that might allow me to copy the moon phase information or share it via email.

<think>Look for a menu or settings icon that might allow copying or sharing the moon phase information.</think>
<answer>click(point='(793, 160)')</answer>

Figure 13: Example from the GUI domain. The multi-agent system receives only the high-level instruction, while the low-level instruction and answer shown here are the ground-truth labels. The red annotations highlight the multi-agent system's errors before interleaved training, while the green annotations show the improved interaction after interleaved training.

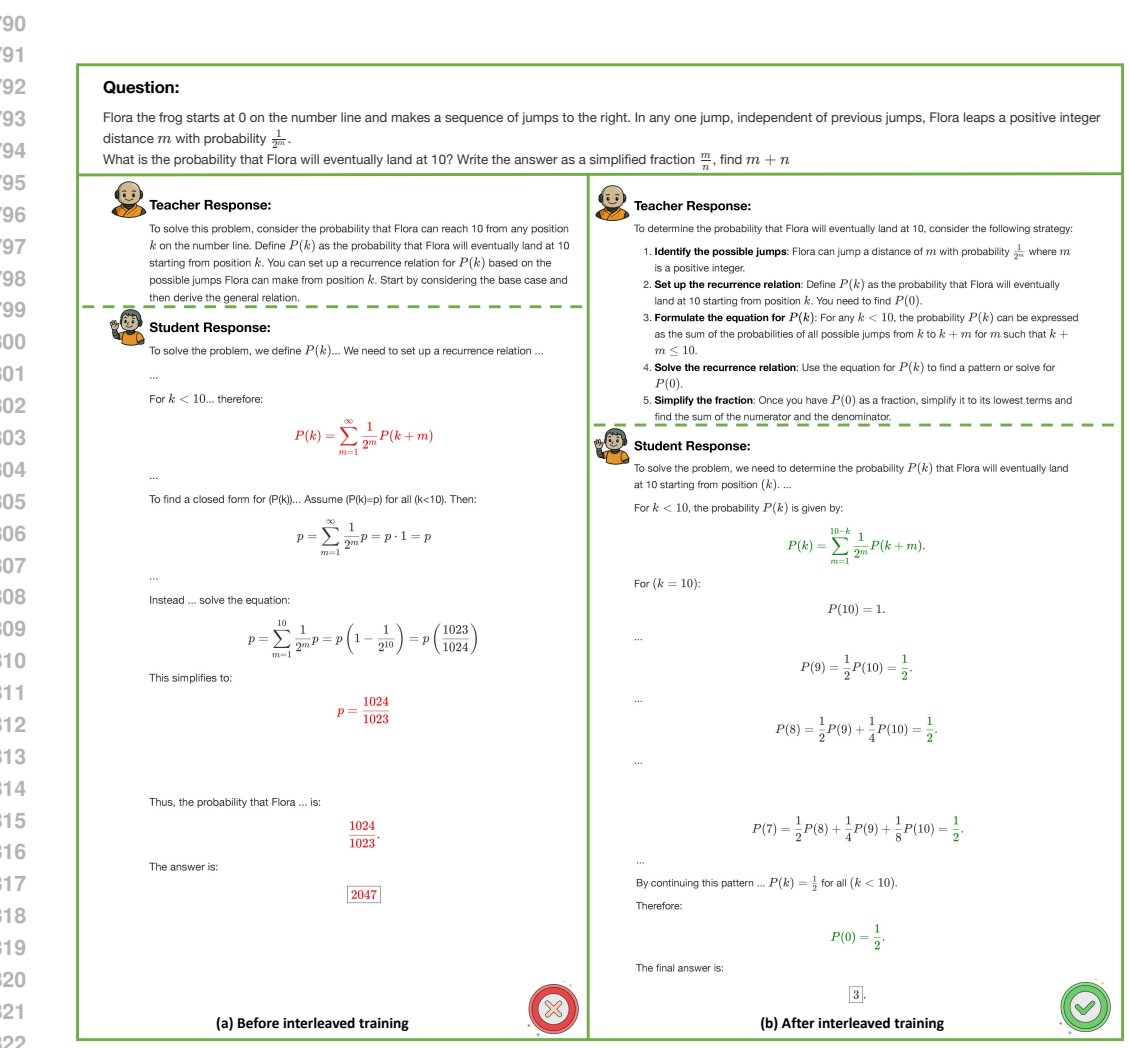

Figure 14: Example from the mathematics domain. The left panel (a) shows the multi-agent system's reasoning before interleaved training, with red annotations highlighting its errors. The right panel (b) shows the system after interleaved training, with green annotations indicating the improved reasoning and final answer.

# I   LIMITATIONS

To isolate the effect of the training scheme from rollout noise or environment stochasticity and to conduct controlled ablations, our experiments focus on static offline GUI datasets rather than dynamic online GUI environments. In addition, our system infrastructure constraints prevented us from validating the method further in fully interactive settings. Moreover, SWIRL trades computation time for memory efficiency: the sequential update mechanism enables $O(1)$ actor memory usage, but this design choice naturally increases the total wall-clock training time because each agent is optimized separately rather than concurrently. Finally, limited training resources prevented us from performing more extensive scaling experiments on substantially larger datasets, and a more systematic exploration of data scaling remains open for future investigation.

