# OpenReview forum: "SWIRL: A Staged Workflow for Interleaved Reinforcement Learning in Mobile GUI Control"
_ICLR.cc/2026/Conference — Submitted to ICLR 2026_

### Official Review · Reviewer_myFk · 2025-10-27

**Soundness:** 3
**Presentation:** 2
**Contribution:** 2
**Rating:** 2
**Confidence:** 4

**Summary:**

This paper introduces SWIRL, a novel RL framework for multi-agent systems.
Specifically, SWIRL conducts training of every single agent sequentially, thus stabilising the entire training process of the multi-agent system.
The authors provide a theoretical analysis for step-wise safety, monotonic improvement, and convergence of SWIRL.
Empirical results on GUI tasks and mathematical reasoning tasks demonstrate the superior performance of SWIRL.

**Strengths:**

1. Different from step-level alternation [1], SWIRL present a novel round-level alternation, sequentializing single-agent updates while maintaining multi-agent coordination.
2. Presenting theoretical results based on [1].
3. Extensive experiments on different benchmarks, including GUI tasks and math reasoning tasks.



>Reference
>>[1] Heterogeneous-agent reinforcement learning.

**Weaknesses:**

1. First concern is the unclear structure, presentation and writing, especially in Section 2. The main contribution is the round-level alternation, while section 2.1 (Lines 59-200) is unclear in illustrating the underlying motivation. Additionally, the theoretical results are based on previous work [1]; it is important to illustrate clearly how to adapt from the previous one to address the challenges in this work (e.g., from step-level to round-level).
2. For GUI tasks, a bunch of related works are missing, including [2-7].
3. Limited novelty. The navigator-interactor pipeline is a common approach in the literature. For example, existing work [6] leverages VLM as the subgoal-generator to provide low-level instructions for the low-level agent.


>Reference
>>
>>[1] Heterogeneous-agent reinforcement learning
>>
>>[2] You only look at screens: Multimodal chain-of-action agents
>>
>>[3] Digirl: Training in-the-wild device-control agents with autonomous reinforcement learning
>>
>>[4] Digi-q: Learning q-value functions for training device-control agents
>>
>>[5] Webrl: Training llm web agents via self-evolving online curriculum reinforcement learning
>>
>>[6] Vsc-rl: Advancing autonomous vision-language agents with variational subgoal-conditioned reinforcement learning
>>
>>[7] Distrl: An asynchronous distributed reinforcement learning framework for on-device control agents

**Questions:**

See Weaknesses.
Additionally, it would be great if more GUI experiments, comparing with existing baselines [1-6] on the AitW benchmark [7].


>Reference
>>
>>[1] You only look at screens: Multimodal chain-of-action agents
>>
>>[2] Digirl: Training in-the-wild device-control agents with autonomous reinforcement learning
>>
>>[3] Digi-q: Learning q-value functions for training device-control agents
>>
>>[4] Webrl: Training llm web agents via self-evolving online curriculum reinforcement learning
>>
>>[5] Vsc-rl: Advancing autonomous vision-language agents with variational subgoal-conditioned reinforcement learning
>>
>>[6] Distrl: An asynchronous distributed reinforcement learning framework for on-device control agents
>>
>>[7] A Large-Scale Dataset for Android Device Control

---

> ### Author Response · Authors · 2025-11-24
> **Response Part [1/3]**
>
> Dear Reviewer myFk,
>
> Thank you for recognizing the novelty of our round-level alternation design, the theoretical foundation, and the breadth of our experiments. We sincerely appreciate your constructive comments and suggestions. Below we provide our detailed responses.
>
> ### **W1.1: The main contribution is the round-level alternation, while section 2.1 is unclear in illustrating the underlying motivation.**
>
> We sincerely thank the reviewer for the careful reading and valuable feedback. **We have substantially rewritten Sec. 2.1 in the revised version to make both the structure and the underlying motivation explicit.** In summary, our round-level alternation scheme is motivated by two complementary perspectives:
>
> **Application perspective.&#x20;**&#x49;n practice, optimizing a large-model-based multi-agent system via standard end-to-end MARL is extremely resource-intensive. It requires loading multiple large backbones simultaneously and maintaining activations, gradients, and optimizer states for all agents, leading to prohibitive memory and compute costs and very low training efficiency. Moreover, existing efficient RL training frameworks for large models are primarily designed for single-agent. This motivates a natural question: can we train the agents in a multi-agent system sequentially, one at a time, so that we achieve O(1) actor memory usage while still optimizing the same cooperative multi-agent objective and fully leveraging mature single-agent RL infrastructure? Based on this practical need, we propose a round-level alternation design: training proceeds in multiple rounds, and within each round, agents are updated sequentially while the others are frozen.
>
> **Theoretical perspective.&#x20;**&#x46;rom an optimization viewpoint, classical alternating-update methods such as ADMM provide an instructive analogy: ADMM improves a coupled global objective by making sufficiently strong progress on one variable at a time, and comes with strong convergence guarantees. Inspired by this principle, we posit an analogous design in our setting: if each per-agent update within a round yields sufficient improvement under fixed teammates, then the overall joint multi-agent policy should also improve across rounds. We follow this line of thought more formally. Specifically, we first present the round-level alternation procedure (Algorithm 1), and then establish three key results: (1) a single-step safety bound, (2) monotonic improvement of the joint return across rounds, and (3) convergence of returns. Together, these results clarify why round-level alternation yields a stable, monotonically improving optimization of the multi-agent objective, and show that our method is not merely an engineering trick but a principled multi-agent alternating-update scheme with explicit theoretical guarantees.
>
> We hope that these revisions make the structure of Sec. 2.1 clearer, sharpen the motivation behind round-level alternation, and fully address your concern.

---

> ### Author Response · Authors · 2025-11-24
> **Response Part [2/3]**
>
> ### **W1.2:  It is important to illustrate clearly how to adapt from the previous theoretical results to address the challenges in this work.**
>
> Thank you for raising this important point and for asking us to clarify how our theoretical results build on and extend previous theoretical results. Our analysis indeed leverages Lemma 6 of \[1], but it is applied at a different level of alternation and is then combined with a new telescoping argument to obtain round-level guarantees.
>
> **How our setting differs.&#x20;**&#x49;n \[1], all agents are updated sequentially within a single step of the algorithm, and Lemma 6 directly bounds the performance change between the old and the new step-level joint policy. In contrast, our Algorithm 1 adopts a round-level interleaving scheme: during round $k$, all other agents $\tau^{-i}\_{k}$ remain fixed while a single agent $i$ undergoes multiple inner updates, $j = 0,\dots,K\_i-1$ before switching to the next agent. Because each agent performs several updates within its inner layer, it is difficult to directly estimate round-level updates.
>
> **How we adapt from step-level to round-level.&#x20;**&#x54;echnically, our adaptation proceeds in two steps:
>
> **(1) Inner-step instantiation.** For each inner update $\pi^{i}\_{k,j} \to \pi^{i}\_{k,j+1}$ with all other agents frozen, we instantiate Lemma 6 of \[1] at the joint policies $\Pi\_{k,i,j}$ and $\Pi\_{k,i,j+1}$. Because only agent $i$ changes, this yields a single-agent surrogate lower bound inequality of the form in Proposition 1.
>
> **(2) Telescoping across inner updates and agents.&#x20;**&#x57;e then sum these inequalities over all inner updates $j = 0,\dots,K\_i-1$ for each agent $i$ and across all agents within round $k$. The intermediate joint return is added in a telescoping fashion, which leads to a **round-level monotonic improvement result**, namely Theorem 1, establishing $J(\pi\_{k+1}) \geq J(\pi\_k)$ across rounds.&#x20;
>
>
>
> To make this connection explicit, **we have added a clarifying paragraph in Appendix C.2 of the revised manuscript that explains**, in words, (i) how Lemma 6 of \[1] is instantiated for each inner update of a single agent under fixed collaborators, and (ii) how summing these bounds over inner updates and agent switches yields the **round-level monotonic improvement theorem** via a telescoping argument. We hope that this clarification and the additional explanations in Appendix C.2 address your concern.
>
>
> \[1] Heterogeneous-agent reinforcement learning: https://arxiv.org/pdf/2304.09870

---

> ### Author Response · Authors · 2025-11-24
> **Response Part [3/3]**
>
> ### **W2 & Q2: Some related works are missing. Additionally, it would be great if comparing with existing baselines on the AitW benchmark.**
>
> Thank you for the valuable suggestions and for pointing us to these important GUI papers. We have revised the manuscript accordingly and added new experiments. Below we summarize the updates.
>
> **(1) Missing GUI-related works.** In the revised manuscript, we have incorporated \[1–6] into the Related Work section with a discussion. All corresponding citations have been added to the bibliography.
>
> **(2) Comparison on AitW.** After carefully reviewing the suggested baselines, we realized that \[2–6] are trained and evaluated in the online, interactive AITW setting, whereas our current pipeline targets the offline, static AITW setting (as in \[1]). These two regimes are not only different in environment assumptions, but also use different evaluation protocols and metrics. As a result, a direct numerical comparison across the dynamic and static settings would be inherently unfair and potentially misleading. We also want to be fully candid that, at this stage, we are unable to run dynamic AITW training and evaluation. Online interaction with LVLM-based multi-agent RL is currently beyond our available engineering and computational resources. We recognize this as a limitation of the present work, and we have now stated it explicitly in the revised manuscript. Therefore, in the revision we focus our AITW comparison on the offline setting, where evaluation is aligned and fair with \[1]. To enable a more comprehensive comparison, we additionally include other offline-capable baselines \[7, 8]. We use the cleaned AITW version \[7], which mitigates redundancy and label noise in the original AitW release. As shown in **Table A**, SWIRL achieves the best performance, even though it is trained on only 3,500 samples rather than the full AITW training set. **These results are included in Appendix Sec. F.8.**
>
>
>
> Table A: Performance on offline AITW.
>
> | Model         | SR        |
> | ------------- | --------- |
> | AUTO-UI       | 34.46     |
> | AUTO-UI+CoAT  | 47.69     |
> | CogAgent      | 44.52     |
> | CogAgent+CoAT | 53.28     |
> | **SWIRL**     | **62.45** |
>
> ---
>
>
> \[1] AutoUI: https://arxiv.org/pdf/2309.11436
>
> \[2] Digirl: https://arxiv.org/pdf/2406.11896
>
> \[3] Digi-q: https://arxiv.org/pdf/2502.15760
>
> \[4] Webrl: https://arxiv.org/pdf/2411.02337
>
> \[5] Vsc-rl: https://arxiv.org/pdf/2502.07949
>
> \[6] Distrl: https://arxiv.org/pdf/2410.14803
>
> \[7] AITZ: https://arxiv.org/pdf/2403.02713
>
> \[8] CogAgent: https://arxiv.org/pdf/2312.08914
>
> ---
>
> ### **W3: Limited novelty. The navigator-interactor pipeline is a common approach in the literature.**
>
> We would like to clarify that our paper **does not aim to claim novelty in proposing a new Navigator–Interactor pipeline for GUI control. Instead, the core contribution of our work lies in a general multi-agent reinforcement learning framework.** The Navigator–Interactor setup is simply one **instantiation** used to evaluate the effectiveness of our training method. Our novelty and contributions are as follows:
>
> **(1) A general multi-agent training framework.&#x20;**&#x57;e introduce SWIRL, which converts MARL into a sequence of single-agent RL subproblems through round-level interleaving. This enables O(1) actor memory usage and naturally supports heterogeneous model sizes, data schedules, and update budgets.
>
> **(2) Formal theoretical guarantees.&#x20;**&#x57;e establish a per-step safety bound, monotonic improvement across rounds, and convergence of the joint return, showing that SWIRL is a principled multi-agent alternating-update method rather than an engineering heuristic.
>
> **(3) A GUI instantiation demonstrating stable and strong performance.&#x20;**&#x42;y applying SWIRL to the Navigator–Interactor architecture for mobile GUI control, we show stable training and strong performance, and comprehensive ablations that clarify when and why interleaving helps.
>
> **(4) Evidence of generality beyond GUI.&#x20;**&#x57;e further apply the exact same training recipe to a non-GUI domain (mathematical reasoning), where SWIRL again yields robust gains on standard benchmarks, highlighting the potential for broader multi-agent applications.

---

### Official Review · Reviewer_dLTb · 2025-10-28

**Soundness:** 3
**Presentation:** 3
**Contribution:** 3
**Rating:** 6
**Confidence:** 3

**Summary:**

The paper introduces SWIRL, a staged, interleaved multi-agent reinforcement learning (MARL) workflow that decomposes multi-agent training into sequential single-agent updates. It presents formal convergence and safety proofs for interleaved updates, empirical validation showing superior zero-shot performance and robustness, and transferability to mathematical reasoning tasks.

**Strengths:**

1. This paper gives a clever recombination of MARL, alternating optimization, and LVLM integration, executed with clear theoretical rigor and practical benefit.
2. Experiments are diverse, spanning GUI control (high-level and low-level) and math reasoning. Baselines include major prior systems (GUI-R1, MARFT, OS-Atlas, ReMA), with clear evidence of superior performance and resource efficiency (O(1) memory scaling).

**Weaknesses:**

1. As a critical stability factor in MARL, interleaving frequency determines how quickly agents co-adapt. This paper does not empirically investigate how interleaving frequency or synchronization cadence affects performance, convergence, or stability.
2. Although SWIRL is compared against OS-Atlas, GUI-R1, and MARFT, it omits recent multi-agent finetuning frameworks such as MultiAgent Debate (Du et al., 2023), AutoGen (Wu et al., 2024a), and Rema (Wan et al., 2025) in full experimental benchmarking. Comparing against them would help position SWIRL within the broader multi-agent LVLM training ecosystem.
3. The paper emphasizes efficiency (O(1) actor memory), but does not quantify training sample efficiency or scaling behavior. The total training set (3,500 samples) is small, yet the authors do not clarify if this is sufficient for convergence or merely a constraint of resources.
4. SWIRL’s performance may be conflated with the capabilities of the underlying Qwen2.5-VL-3B models. The absence of experiments with smaller or weaker backbones makes it difficult to isolate what proportion of improvement derives from the interleaved RL algorithm itself.
5. The mathematical reasoning experiments (Section 3.4) show numerical improvements but do not explain why interleaving helps reasoning tasks or provide qualitative examples of improvement.

**Questions:**

See the Weaknesses Section.

---

> ### Author Response · Authors · 2025-11-24
> **Response Part [1/4]**
>
> Dear Reviewer dLTb,
>
> Thank you for the overall positive assessment of our work, and for recognizing the clear theoretical rigor, the practical advantages of our alternating multi-agent training design, and the breadth and strength of our experimental evaluations. We sincerely appreciate your constructive comments and suggestions. Below, we provide detailed responses to address the concerns you raised.
>
> ### **W1: This paper does not empirically investigate how interleaving frequency or synchronization cadence affects performance, convergence, or stability.**
>
> We agree that (i) how often agents alternate their updates and (ii) how deeply each agent is trained before switching are key factors governing co-adaptation and stable optimization. **We would like to note that the original submission already included systematic experiments along both dimensions**, and we summarize these results here for clarity.
>
> **(1) Interleaving frequency versus training depth.&#x20;**&#x54;o quantify the influence of alternation frequency and single-agent training depth, we vary the number of alternating rounds (frequency of coordination) and the number of epochs per round (depth of per-agent optimization), while keeping the total training budget fixed. As shown in **Fig. 5(a) and Table 11**, **increasing the number of alternating rounds consistently produces larger performance gains than using the same budget for deeper single-agent training with fewer rounds**. Across all configurations, the **learning curves increase smoothly** without signs of saturation or instability.
>
> **(2) Effect of deeper per-agent training.&#x20;**&#x57;e further investigate whether increasing training depth alone remains beneficial. **Fig. 5(b)&#x20;**&#x73;hows that **deeper single-agent optimization within each round continues to yield incremental improvements**, indicating that per-agent refinement still contributes positively even when the interleaving frequency is fixed.
>
> **(3) Conclusion.&#x20;**&#x54;aken together, these results show that, under a fixed training budget, **more frequent inter-agent coordination is the primary driver of performance improvement**, while **deeper per-agent training plays a complementary but still meaningful role**. In all cases, the **optimization remains stable**, with no evidence of divergence or performance collapse.
>
> ---
>
> ### **W2: It omits recent multi-agent finetuning frameworks such as MultiAgent Debate, AutoGen, and Rema.**
>
> **We would like to kindly point out that the original submission already included a comparison with REMA in Table 4 of Sec. 3.4**. For your convenience, we reproduce the results in **Table A**. Under the same model architecture and the same training data, our method achieves the best average performance across multiple benchmarks.
>
>
>
> Following your suggestion, we have additionally included comparisons with MultiAgent Debate and AutoGen, although they are not multi-agent finetuning frameworks in the same sense as REMA. As described i&#x6E;**&#x20;Sec. 3.4,** our math experiments construct multi-agent systems using Qwen2.5-3B-Coder and Qwen2.5-7B as base models, trained with SWIRL (denoted as SWIRL-3B and SWIRL-7B, respectively).
> The results are summarized in **Table B**, where SWIRL-7B achieves the strongest overall performance among all methods.
>
> ---
>
> Table A: Math reasoning performance with ReMA and SWIRL.
>
> | Method | MATH500 | GSM8K | Minerva | OlympiadBench | Gaokao2023en | AMC23 | AIME24 | Overall |
> | ------ | ------- | ----- | ------- | ------------- | ------------ | ----- | ------ | ------- |
> | ReMA   | 74.4    | 90.6  | 34.9    | 36.3          | 57.9         | 57.5  | 20.0   | 53.1    |
> | SWIRL  | 74.4    | 91.9  | 36.0    | 41.9          | 64.9         | 57.5  | 10.0   | 53.8    |
> ---
>
> Table B: Math reasoning performance with SWIRL, MultiAgent Debate and AutoGen.
>
> | Multi-Agent System | Method        | Base model       | GSM8K | MATH  |
> | ------------------ | ------------- | ---------------- | ----- | ----- |
> | MultiAgent Debate  | training-free | GPT-3.5          | 85.0  | -     |
> | AutoGen            | training-free | GPT-4            | -     | 69.48 |
> | SWIRL-3B           | SWIRL         | Qwen2.5-3B-Coder | 81.4  | 63.94 |
> | SWIRL-7B           | SWIRL         | Qwen2.5-7B       | 91.9  | 73.98 |
> ---

---

> ### Author Response · Authors · 2025-11-24
> **Response Part [2/4]**
>
> ### **W3: Does not quantify the efficiency or scaling of the training data.**
>
> We thank the reviewer for pointing out the lack of explicit analysis of training sample efficiency and scaling behavior.&#x20;
>
> **(1) About scaling of the training data.&#x20;**&#x57;e agree that a more systematic study on data scaling would further strengthen the paper. Our current experiments use a relatively small dataset of 3,500 samples, comparable to prior RL-based LVLM GUI-control work such as GUI-R1, and this choice is **driven by computational limitations**: large-scale LVLM multi-agent RL is extremely costly to run. Unfortunately, we are currently unable to support a comprehensive large-scale study. We appreciate the reviewer for highlighting this point and **have added an explicit statement in the "Limitations" section (Appendix Sec. I)** to clarify this constraint.
>
> **(2) About the efficiency of the training data.** To ensure effective training under resource constraints, we designed two methods to improve sample efficiency:
>
> **High-quality data selection pipeline.** Before training, we apply a strict filtering pipeline (**Appendix Sec. E.1**) to select high-quality samples and discard noisy or low-signal data. This is intended to ensure that each of the 3,500 retained samples provides a meaningful learning signal.
>
> **Online reweighting for efficient learning.** During training, we employ an online reweighting mechanism (**Sec 2.3**) that lets the model adaptively focus on more informative samples. This helps the agents make more effective use of the available data budget and improves training efficiency in the resource-limited setting. We also provide a detailed quantitative analysis of the effect of online reweighting in the **Appendix F.4**. It indicates that the online reweighting mechanism progressively down-weights low-information "easy" samples and prioritizes harder, more informative ones, thereby improving training efficiency and avoiding over-emphasis on uninformative or potentially misleading data.

---

> ### Author Response · Authors · 2025-11-24
> **Response Part [3/4]**
>
> ### **W4: SWIRL's performance may be conflated with the capabilities of the underlying Qwen2.5-VL-3B models.**
>
> **Effect across different backbone capacities.&#x20;**&#x49;n the math experiments, we deliberately used two backbones with different parameter scales **(i.e., Qwen2.5-3B-Coder and Qwen2.5-7B)** to examine whether SWIRL's gains depend on model capacity. For your convenience, we reproduce the results in **Table C and Table D**. We observe consistent improvements on both model sizes after applying SWIRL, and the gains are in fact larger for the smaller 3B model (an increase of +10.3 overall points). This suggests tha&#x74;**&#x20;the performance improvements are not attributable to backbone strength alone; if anything, SWIRL appears to provide greater benefit to weaker models.**
>
> **Effectiveness of SWIRL is independent of backbone capability. Sec. 3.5** includes an ablation that isolates the contribution of the interleaved update scheme itself. Using the same backbone and the same training data, we compare SWIRL with an isolated training pipeline where the Navigator is trained first and the Interactor is then trained on top of the frozen Navigator, and we implement this pipeline in both SFT and RL variants. We reproduce these results in **Table E**. SWIRL clearly outperforms all isolated-training variants, demonstrating that th&#x65;**&#x20;improvements arise from the interleaved RL algorithm rather than from the underlying model capacity.**
>
> **Fair comparison under controlled training conditions.&#x20;**&#x54;o further disentangle model capability from training method, the math experiments in **Sec. 3.4** follow the exact backbone and data setups used in prior multi-agent training frameworks (e.g., Qwen2.5-3B-Coder for MARFT and Qwen2.5-7B for REMA). In this setting, **the only variable is the training algorithm**. As shown again in **Table C and Table D,** SWIRL achieves superior performance under these controlled conditions, highlighting that the **gains are attributable to the interleaved RL strategy itself rather than to differences in backbone strength.**
>
> Table C: Math reasoning performance on **Qwen2.5-7B&#x20;**&#x75;nder different training and agent configurations.
>
> | Method | Base model | MATH500  | GSM8K    | Minerva  | OlympiadBench | Gaokao2023en | AMC23    | AIME24   | Overall  |
> | ------ | ---------- | -------- | -------- | -------- | ------------- | ------------ | -------- | -------- | -------- |
> | SOLO   | Qwen2.5-7B | **75.0** | **92.0** | 35.7     | 38.2          | 56.6         | 47.5     | 6.7      | 50.2     |
> | ReMA   | Qwen2.5-7B | 74.4     | 90.6     | 34.9     | 36.3          | 57.9         | **57.5** | **20.0** | 53.1     |
> | SWIRL  | Qwen2.5-7B | 74.4     | 91.9     | **36.0** | **41.9**      | **64.9**     | **57.5** | 10.0     | **53.8** |
>
> ---
>
> Table D: Math reasoning performance on **Qwen2.5-3B-Coder** under different training and agent configurations.
>
> | Method | Base model       | MATH500  | CMATH    | GSM8K    | Overall  |
> | ------ | ---------------- | -------- | -------- | -------- | -------- |
> | SOLO   | Qwen2.5-3B-Coder | 40.4     | 81.4     | 76.8     | 66.2     |
> | MARFT  | Qwen2.5-3B-Coder | 49.8     | 83.0     | 78.7     | 70.5     |
> | SWIRL  | Qwen2.5-3B-Coder | **64.6** | **83.5** | **81.4** | **76.5** |
>
> ---
>
> Table E:  Comparison Between Isolated training and Interleaved Training (SWIRL).
>
> | Training Strategy              | AndroidControl    |       AndroidControl            |        AndroidControl           | GUI-Odyssey       |          GUI-Odyssey         |          GUI-Odyssey          | overall           |
> | ------------------------------ | ----------------- | ----------------- | ----------------- | ----------------- | ----------------- | ----------------- | ----------------- |
> |                                | Type              | GR                | SR                | Type              | GR                | SR                |                   |
> | Baseline                       | 62.87             | 69.00             | 46.83             | 70.73             | 63.32             | 46.42             | 60.01             |
> | + Isolated SFT                 | 63.77 **(+0.90)** | 70.25 **(+1.25)** | 48.04 **(+1.21)** | 72.25 **(+1.52)** | 66.19 **(+2.87)** | 49.64 **(+3.22)** | 61.69 **(+1.68)** |
> | + Isolated RL                  | 64.87 **(+2.00)** | 70.68 **(+1.68)** | 49.34 **(+2.51)** | 72.01 **(+1.28)** | 63.98 **(+0.66)** | 47.97 **(+1.55)** | 61.48 **(+1.47)** |
> | + Interleaved Learning (SWIRL) | **66.72 (+3.85)** | **71.19 (+1.29)** | **51.24 (+4.41)** | **74.87 (+4.14)** | **66.39 (+3.07)** | **51.65 (+5.23)** | **63.68 (+3.67)** |
> ---

---

> ### Author Response · Authors · 2025-11-24
> **Response Part [4/4]**
>
> ### **W5: Do not explain why interleaving helps reasoning tasks or provide qualitative examples of improvement.**
>
> We thank the reviewer for pointing this out. **We have added a concrete qualitative example to the paper (see new Figure 14 in the Appendix H)&#x20;**&#x61;nd used it to clarify why interleaving helps on reasoning tasks.
>
> **Qualitative example.&#x20;**&#x57;e illustrate the effect of SWIRL with a representative math reasoning example in **Fig. 14.** On the **left** (before interleaved training), the teacher's hint is extremely minimal and does not clearly identify the key steps of the solution, so the student follows a confused reasoning path and arrives at an incorrect answer. On the **right** (after SWIRL training), the teacher's hint is precise, targeted, and well organized: it highlights the crucial intermediate steps and clarifies the overall structure of the solution. The student is then able to follow this guidance and solve the problem correctly. This side-by-side comparison shows that SWIRL training leads to better coordination between teacher and student: **the teacher learns to give hints that genuinely help the student solve the task, and the student's own reasoning ability improves under these more informative hints.**
>
> **Why interleaving helps reasoning tasks.&#x20;**&#x54;his example is representative of the general mechanism behind the quantitative gains reported in Sec. 3.4. In our setting, the teacher is responsible for generating hints or intermediate guidance, and the student is responsible for solving the full problem. If the teacher produces generic or poorly structured hints, and the student is not specifically adapted to make use of those hints, then the overall performance is naturally poor. Our SWIRL interleaved training explicitly encourages the two models to adapt and align with each other: the teacher is updated using feedback from the student's final correctness, while the student is updated conditioned on the teacher's current hint&#x73;**.** This encourages:
>
> * **The teacher to generate more focused and structured hints:&#x20;**&#x69;nstead of poorly structured suggestions, the teacher learns to produce clear, stepwise guidance that directly targets the main reasoning bottlenecks faced by the student.
>
> * **The student to better exploit the teacher's hints:** because the student is trained under these evolving hints, it becomes increasingly capable of following the hinted decomposition and turning it into a correct multi-step solution.
>
> Together, this co-adaptation explains why interleaving improves performance on mathematical reasoning tasks and is consistent with the qualitative behavior exhibited in **Fig. 14**.

---

### Official Review · Reviewer_Qt5R · 2025-10-29

**Soundness:** 3
**Presentation:** 2
**Contribution:** 2
**Rating:** 2
**Confidence:** 4

**Summary:**

This paper introduces SWIRL, a method that frames a MARL problem but leverages single-agent algorithms through sequential freezing and interleaving of agents during training. Agents included are Navigator and Interactor, dealing with UI screenshots and user requests as well as low level instructions from the Navigator and UI screenshots to derive actions (Interactor). Authors claim for this setup to outperform SOTA models such as GPT5, R1 etc.

**Strengths:**

- Interesting idea
- Good experiments on replacing the Navigator with SOTA models such as GPT5 etc. to get  Interactor performance

**Weaknesses:**

- It is not clear to me why this is formulated as a multi-agent problem, as the setup is stated as a "...sequential single-agent... enabling reuse of standard RL..."

- I believe the main weakness of the paper is the experiment design to answer the question above. In the experiment, I would have expected the following experiments:
1. End-to-end RL training of Navigator and Interactor, potentially sharing rewards
2. Isolated training of the Navigator until convergence and then isolated training of the Interactor (with frozen, well-performing Navigator)
3. Interleafed Navigator / Interactor (SWIRL, yours) training

This is in combination with my main concern: The Indicator is clearly dependent on the performance of the Navigator. If the Navigator performs badly, the Interactor has no chance of performing well.

- I don't see clear conclusion and discussion why the interleafed setup performs better than all other setups.

**Questions:**

Why are you reporting performance on pre- and post-training language benchmarks (GSM8K, MATH500 etc.) when the paper is looking at vision-language models?

I would be interested in the isolated performance of the Navigator and the Interactor. Why did you decide not to show that?

---

> ### Author Response · Authors · 2025-11-24
> **Response Part [1/5]**
>
> Dear Reviewer Qt5R,
>
>
> Thank you for recognizing the novelty of our idea and our experimental design. We sincerely appreciate your constructive comments and suggestions. Below we provide our detailed responses to address your concerns.
>
> ### **W1: Not clear why this is formulated as a multi-agent problem**
>
> We thank the reviewer for the insightful comments. We understand that the previous phrasing may inadvertently create the wrong impression. To clarify, **the problem formulation, learning objective, and theoretical results are intrinsically and unequivocally multi-agent. The sequential updates are merely a computational strategy and do not alter the fundamentally multi-agent nature of the framework**. We have revised the paper to make this distinction clearer.
>
> **(1) Why the problem is formulated as a multi-agent problem.&#x20;**&#x4F;ur GUI control setting comprises two distinct decision-making entities: a **navigator** that produces low-level instructions and an **interactor** that executes concrete GUI actions. Each entity has its **own policy**, **action interface**, and **role** in the decision process. The task outcome and reward depend on the composition of their behaviors: the navigator's instruction determines the interactor's action trajectory, and success is evaluated on the final GUI execution. Hence the return is a function of the **joint policy** $(\pi\_{\text{nav}}, \pi\_{\text{int}})$. This is exactly the multi-agent setting: **multiple policies jointly determine performance under a shared reward**. Therefore, the optimization target is genuinely multi-agent. Moreover, our theoretical analysis is explicitly conducted under this multi-agent formulation. Proposition 1, Theorem 1, and Corollary 1 are all stated in terms of the joint return achieved by the multi-agent policy produced via SWIRL's interleaved learning.&#x20;
>
> **(2) What we mean by "sequential single-agent … enabling reuse of standard RL."&#x20;**&#x53;WIRL does **not** re-cast the task as a single-agent MDP. Instead, **it solves the multi-agent objective $J(\pi\_{\text{nav}}, \pi\_{\text{int}})$ using round-wise alternating updates.** In each round, SWIRL: 1) holds the other agent's policy fixed, 2) optimizes the active agent in the induced environment, and 3) switches to the next agent. Thus, although each update step is a single-agent RL subproblem conditioned on the fixed policies of the other agents, over the course of a full round, all agents are updated in turn. The overall optimization is decomposed into per-agent single-agent update subproblems, while the underlying environment and objective remain multi-agent. **The sequential single-agent view is therefore a computational decomposition, not a change of problem class: the environment and objective remain multi-agent, and the learned solution is a multi-agent joint policy.**
>
> **(3) Motivation for the alternating scheme.&#x20;**&#x41; direct (simultaneous) MARL training pipeline for this multi-agent system would require all agents to remain train-active throughout optimization (i.e., loading every policy concurrently and maintaining their optimizer states and gradient graphs during backpropagation). In large model settings, this leads to substantial peak memory and compute overhead, which makes naive simultaneous MARL impractical. **Precisely because of this overhead**, we adopt such a round-wise alternating scheme. This alternating procedure preserves the original multi-agent objective while making training feasible and allowing the use of standard single-agent RL solvers as conditional subproblem optimizers. Importantly,**&#x20;these "sequential single-agent updates" are merely a technical tool for solving the per-agent subproblems under fixed teammates, and do not reduce the underlying formulation or objective to a single-agent setting.**
>
> **(4) Revision of the potentially misleading sentence.&#x20;**&#x57;e have revised the potentially misleading sentence **(around line 100)** to make the above distinction explicit:
>
> > "…turning joint optimization into sequential single-agent problems and enabling reuse of standard RL…"
>
> with:
>
> > Unlike traditional concurrent MARL, where all agents are updated simultaneously, SWIRL optimizes the multi-agent objective via round-wise alternating updates: each agent is improved in turn using a standard single-agent RL solver while the other agent's policy is held fixed; over a full round, all agents are updated sequentially. Single-agent RL is used purely as a subproblem solver; the environment and objective remain multi-agent, and all guarantees are stated for the resulting joint policy.
>
> We thank the reviewer again for highlighting this ambiguity, which helped us significantly improve the clarity and precision of our presentation.

---

> ### Author Response · Authors · 2025-11-24
> **Response Part [2/5]**
>
> ### **W2.1: Expected experiment design across the three suggested setups.**
>
>
> We appreciate your valuable suggestions.
>
> Regarding experiment (1), fully end-to-end RL training of both the Navigator and Interactor is unfortunately not feasible in our setting. As discussed in our response to W1, a direct MARL-style end-to-end update would require keeping all large models simultaneously train-active, leading to peak memory and compute demands that **far exceed the resources available to us**. Moreover, current community tools and techniques do not yet provide scalable support for end-to-end training of large-model multi-agent systems at this scale. **This is precisely why SWIRL is designed the way it is:** instead of abandoning the multi-agent formulation, we convert the intractable joint optimization problem into tractable round-wise alternating updates. This design preserves the multi-agent objective while making the training process realistically executable with available resources.
>
> Regarding experiment (2), **we note that our original Sec. 3.5 already included an isolated-training baseline in the form of SFT-based training of the multi-agent system.** We apologize if the presentation may have caused confusion. In addition to the SFT variant, we have now **added an RL-based isolated-training baseline as well**. As shown in **Table A**, interleaved training in SWIRL outperforms training the agents in isolation. **The updated manuscript now includes these results in Sec. 3.5.** We have also revised the previously complex and hard-to-follow description of the experimental setup to present the results more clearly and transparently. In addition, the following Section W2.2 of the rebuttal provides a detailed analysis of why interleaved training is more effective than isolated training in this cooperative multi-agent context.
>
> Table A: Comparison Between Isolated and Interleaved Training.
>
> | Training method              | AndroidControl |     AndroidControl      |     AndroidControl     | GUI-Odyssey |    GUI-Odyssey       |     GUI-Odyssey      | overall   |
> | ---------------------------- | -------------- | --------- | --------- | ---------- | --------- | --------- | --------- |
> |                              | Type           | GR        | SR        | Type       | GR        | SR        |           |
> | Isolated RL                  | 64.87          | 70.68     | 49.34     | 72.01      | 63.98     | 47.97     | 61.48     |
> | Isolated SFT                 | 63.77          | 70.25     | 48.04     | 72.25      | 66.19     | 49.64     | 61.69     |
> | Interleaved Learning (SWIRL) | **66.72**      | **71.19** | **51.24** | **74.87**  | **66.39** | **51.65** | **63.68** |
>
> ---

---

> ### Author Response · Authors · 2025-11-24
> **Response Part [3/5]**
>
> ### **W 2.2: Why the interleaved setup performs better than all other setups.**
>
> We thank the reviewer for raising this point. The previous version already included an analysis of the interleaved scheme; however, to make the central takeaway more accessible, we provide a clearer and more focused articulation of it here, supported by both theoretical guarantees and empirical evidence. In short, **the interleaved scheme provides a provably stable and monotonically improving optimization process, and under a comparable training budget, it yields stronger, better-aligned agents whose repeated realignment through alternation leads to superior collaborative performance of the overall multi-agent system.** We summarize the supporting arguments below.
>
> **(1) Theoretical justification for the interleaved scheme.&#x20;**&#x4F;ur theoretical results, including Theorem 1 and Corollary 1, provide guarantees for the multi-agent policy optimized via interleaved updating. They show that, under the round-wise interleaved schedule, the joint return is monotonically improving across rounds and the sequence of returns converges. These results justify the use of interleaved update as a sound optimization strategy in multi-agent settings: **they ensure that the overall system enjoys a stable, monotonically improving, and convergent optimization trajectory**. Consistent with these guarantees, **Fig. 4** in the Appendix empirically demonstrates that performance indeed improves steadily during training.
>
> **(2) Empirical evidence supporting the effect of interleaved updating.&#x20;**(i) As shown in the additional ablations in **Appendix Sec. F.1**, interleaved training not only improves the joint multi-agent policy but also **strengthens the capability of each individual agent.** Because updates alternate across agents, each agent repeatedly improves while operating under a stable snapshot of its partner. This allows the Navigator and Interactor to deepen their role-specific specialization (i.e., the Navigator becomes more proficient at decomposing complex tasks, while the Interactor becomes increasingly skilled at executing low-level GUI actions). This role-focused skill consolidation, which is absent in isolated training, **enables each agent to better fulfill its designated function within the overall system**. (ii) **Table A** shows that interleaved training consistently outperforms isolated training across both tasks and metrics. Conceptually,**&#x20;isolated training can be viewed as a degenerate case of interleaving in which only a single round of updates is performed (i.e., round = 1).&#x20;**(iii) Furthermore, **Appendix Sec. F.3&#x20;**&#x73;hows that, when the total training budget is fixed, increasing the number of alternation rounds consistently improves coordination between agents.**&#x20;More frequent alternations provide additional opportunities for the Navigator and Interactor to adapt to each other's updated policies, leading to repeated realignment.** As a result, each agent becomes better calibrated to its partner's output characteristics (i.e., it "understands" its partner better), resulting in stronger cooperative dynamics and improved overall performance.

---

> ### Author Response · Authors · 2025-11-24
> **Response Part [4/5]**
>
> ### **W3.  The Indicator is clearly dependent on the performance of the Navigator. If the Navigator performs badly, the Interactor has no chance of performing well.**
>
> We thank the reviewer for raising this concern. The Interactor is designed to condition on the Navigator's outputs, and we did not intend to suggest that our method removes this dependency. However, we empirically observe that **SWIRL training mitigates how this dependency manifests in practice**. The new qualitative example in **Fig. 13 of Appendix H&#x20;**&#x20;shows that, for more challenging tasks, the trained Navigator tends to **(i) exhibit a deeper global understanding of the task** and **(ii) provide less over-committed instructions rather than rigid low-level actions**. Before training, the Navigator often commits prematurely to specific actions; when these are wrong, the Interactor is forced to follow a mistaken path, which matches the failure mode highlighted by the reviewer. After training, the Navigator instead offers softer, exploration-friendly guidance (e.g., goals and possible strategies), leaving the Interactor room to adapt and correct minor mistakes.
>
> In this sense, **the** **Navigator learns to better coordinate with the Interactor**, and **the** **Interactor becomes more capable of interpreting and refining the Navigator's hints**, which is **consistent with the co-evolution behavior** we observe in Appendix F.1. While this does not eliminate the dependency, it helps alleviate brittle failures caused by a single incorrect low-level instruction.&#x20;

---

> ### Author Response · Authors · 2025-11-24
> **Response Part [5/5]**
>
> ### **Q1. Why are you reporting performance on language benchmarks when the paper is looking at vision-language models?**
>
> The reason we also report results on language benchmarks is **not** to shift the focus away from vision-language models, but **to evaluate the generality of SWIRL**. For a fair and direct comparison with prior multi-agent training frameworks such as REMA and MARFT, we **follow their experimental setups and use the same language-based mathematical reasoning benchmarks**. Results show that SWIRL also brings consistent improvements in settings beyond GUI control, underscoring its robustness and broad applicability. The language-only experiments are included purely as supplementary evidence to demonstrate that SWIRL's benefits are not confined to a single modality or task family.
>
> We also emphasize that **SWIRL is inherently modality-agnostic**: the method only assumes a cooperative multi-agent setup optimized via staged, interleaved RL, and does not rely on the nature of the input modality. In our GUI experiments, the agents are LVLMs operating on mobile screens; in the math experiments, they are text-only reasoning agents. Since SWIRL operates on the training paradigm rather than the input format, including language benchmarks simply provides additional confirmation that the algorithm generalizes across modalities without detracting from the paper's central contribution.
>
> ---
>
> ### **Q2. I would be interested in the isolated performance of the Navigator and the Interactor. Why did you decide not to show that?**
>
> **We would like to kindly point out that the original submission already included part of the isolated evaluation, and we have now further expanded these results to address your concerns more comprehensively.** For convenience, we summarize both the previously reported and newly added evaluations below:
>
> **(1) Interactor in isolation.&#x20;**&#x42;ecause the Interactor directly produces executable GUI actions, we evaluate its isolated capability using the low-level action benchmarks i**n Sec. 3.3**. As shown **in Table 3 and Table 5**, the Interactor trained with our interleaved scheme achieves the strongest performance across these benchmarks, demonstrating that SWIRL yields a highly capable Interactor even when evaluated in isolation.
>
> **(2) Navigator in isolation.&#x20;**&#x57;e include an evaluation of the Navigator in isolation, shown in **Table B.** Since the Navigator converts coarse high-level instructions into fine-grained low-level instructions (e.g., "open the Chrome app"), its outputs are textual and cannot be reliably judged by string-matching metrics alone. We therefore report three complementary indicators: **(i)F1 score** against ground-truth instructions, **(ii) Human evaluation** on 200 sampled cases (100 per dataset), where model-blind annotators assign a 0–1 alignment score, **(iii)Execution-based evaluation**, where we assess whether the strongest low-level-instruction executor (i.e., our Interactor), can successfully execute the Navigator's generated instruction.
>
> Results indicate that although our Navigator is not always closest to the ground truth by human-evaluated textual metrics, it consistently leads to higher execution success when paired with the Interactor. This indicates that in GUI control, the Navigator **is not the bottleneck**; task success depends on **cooperation between agents** rather than the isolated quality of one of them. This also further addresses the concern raised in W3 regarding the upper-bound of the Navigator.**&#x20;We have incorporated all these new results into the revised manuscript (Appendix Sec. F.7).**
>
> **(3) Evolution of isolated Navigator and Interactor during training. Appendix Sec. F.1 (Table 9 and Table 10)&#x20;**&#x66;urther shows that both the Navigator and Interactor substantially improve over the course of interleaved training. These results reinforce that SWIRL's alternating update procedure **strengthens the capability of each individual agent**.
>
> Table B: The isolated performance of Navigator.
>
> | model          | AndriodControl |    AndriodControl        |        AndriodControl       | GUI-Odyssey |      GUI-Odyssey      |       GUI-Odyssey        |
> | -------------- | -------------- | ---------- | ------------- | ----------- | ---------- | ------------- |
> |                | F1             | human-eval | Interactor SR | F1          | human-eval | Interactor SR |
> | Qwen2.5-VL-3B  | 19.75          | 0.44       | 44.50         | 20.18       | 0.36       | 40.14         |
> | Qwen2.5-VL-7B  | 20.92          | 0.46       | 47.70         | 22.72       | **0.50**   | 43.84         |
> | UI-Tars-1.5-7B | 15.80          | 0.52       | 49.30         | 13.96       | **0.50**   | 48.56         |
> | GPT-5          | 30.71          | **0.56**   | 49.53         | 23.79       | 0.35       | 44.21         |
> | Navigator      | **36.65**      | 0.54       | **51.24**     | **37.56**   | 0.36       | **51.65**     |
>
> ---

---

### Official Review · Reviewer_t83B · 2025-10-30

**Soundness:** 4
**Presentation:** 4
**Contribution:** 3
**Rating:** 4
**Confidence:** 2

**Summary:**

This paper introduces SWIRL, a framework for reformulating MARL as a sequence of single-agent tasks, therefore sidestepping many of the known challenges, such as stability. The authors provide theoretical guarantees including per-step safety bounds, monotonic improvement across rounds, and convergence properties. In particular, this paper focuses on improving the capabilities of mobile GUI agents, while also testing on mathematical benchmarks. The paper achieves very strong empirical results.

**Strengths:**

**Impressive empirical results** - SWIRL outperforms or achieves near state-of-the-art in a variety of benchmarks. This is a major strength of the paper.

**Problem is important** - Mobile GUI control is a useful area of research, so I believe the paper is well-motivated and will be of use to the community.

**Transfer beyond GUI** - I appreciate that this paper extends its results to more than one benchmark, namely mathematical benchmarks. Validity of the work is improved by showing results are not just closely tied to one benchmark.

**Source Code** - The source code being included in the supplementary material improves the validity and reproducibility of this work.

**Weaknesses:**

- **Support for $>2$ agents** - While the proposed SWIRL framework is reported to support N different agents, all experiments focus on settings with only two agents. Do you have any results demonstrating a setting with more than 2 agents?

- **Walltime** - Currently, the paper does not disclose any information about the wall time of this method. I would imagine that the alternating nature of the approach may increase this substantially.

- **Limitations Section?** - There appears to be no limitations section in this paper. This is a considerable red flag, as this is almost always an important section of research papers.

- **Wording of Claims** - From the wording of this paper (such as line 20), it seems as though the idea of interleaved training with fixed agents is novel. While the authors acknowledge that this has been done before (via citing numerous prior works), I still feel that the authors should be careful not to overreach in terms of their claims.

**Questions:**

- How does this approach work when scaling to more than 2 agents?

- What impact does this approach have on walltime?

- Where is the limitations section of this paper? What are the limitations?

I will cautiously provide a score of 4, given that this is not my primary area of expertise; however, I will raise my score if my concerns are addressed.

---

> ### Author Response · Authors · 2025-11-24
> **Response Part [1/2]**
>
> Dear Reviewer t83B,
>
> Thank you for recognizing our work's empirical strength, transferability, and reproducibility. Your constructive comments and suggestions are valuable to us. Below we provide our detailed responses to address your concerns.
>
> ### **W1\&Q1: Do you have any results demonstrating a setting with more than 2 agents?**
>
> Thank you for pointing this out. **In the revised manuscript (Appendix Sec. F.6), we now include a three-agent system experiment**:
>
> We add a **Summarizer** agent in front of the Navigator–Interactor pair.&#x20;
>
> At each step, the Summarizer converts up to four historical screenshots into a concise textual description summarizing the GUI trajectory, which is then provided as input to the Navigator. The Navigator and Interactor keep the same roles as in the main paper (i.e., the Navigator produces low-level instructions, and the Interactor executes them as atomic GUI actions).
>
> Training follows exactly the SWIRL recipe: we alternate RL updates over the three agents in the order **Summarizer → Navigator → Interactor**, keeping the other agents frozen when one is being updated. Thanks to SWIRL's O(1) actor memory footprint, we reuse the same training hardware as in the 2-agent experiments and do not add extra devices (i.e.,  8 A800 GPUs for training the currently updated agent and another 8 A800 GPUs for deploying the remaining two agents.).
>
> As shown in **Table A**, SWIRL improves the overall average score from 53.56 to 62.79 (+9.23 points), with consistent gains across all three high-level benchmarks. We believe this new experiment empirically demonstrates that SWIRL scales beyond two agents and supports more agent architectures in practice, not only in theory.
>
> Table A: Three-agent performance on in-domain (AITZ) and out-of-domain (AndroidControl, GUI-Odyssey) GUI benchmarks.
>
> |          | AITZ |    AITZ   |   AITZ    | AndroidControl  |   AndroidControl    |   AndroidControl    | GUI-Odyssey |     GUI-Odyssey  |   GUI-Odyssey    | Overall |
> | -------- | ---------------- | ----- | ----- | ------------------------------ | ----- | ----- | --------------------------- | ----- | ----- | ------- |
> |          | TYPE             | GR    | SR    | TYPE                           | GR    | SR    | TYPE                        | GR    | SR    |         |
> | Baseline | 55.25            | 83.01 | 41.98 | 54.80                          | 58.59 | 33.12 | 62.53                       | 57.49 | 35.23 | 53.56   |
> | SWIRL    | 60.75            | 89.34 | 53.26 | 61.42                          | 70.34 | 45.46 | 71.20                       | 65.53 | 47.82 | 62.79   |
>
> ---
>
> ### **W2\&Q2: Does not disclose any information about the wall time of this method.**
>
> Thank you for raising this point. **In the revised manuscript (Appendix Sec. E.1), we now report detailed wall-clock times for the SWIRL training procedure.&#x20;**
>
> Under our main experimental setup (20 rounds), the end-to-end wall time is approximately 85 hours. We also summarize the per-round training time for different numbers of agents in **Table B**. Concretely, for the two-agent setting, one round of SWIRL training takes roughly **2.5×** the wall time of training a single model. This reflects the expected trade-off between training cost and multi-agent coordination enabled by alternating updates.
>
> Table B: Per-round training time under different numbers of agents.
>
> | # Agents | Agent Training Time |  Other Overheads *(e.g., data generation, agent deployment, initialization)*   | Total Time |
> | -------- | ------------------- | ---------------------------------------------------------------------------- | ---------- |
> | 1        | 95min               | 10min                                                                          | 105min     |
> | 2        | 140 min + 80 min   | 35min                                                                        | 255min     |
> ---

---

> ### Author Response · Authors · 2025-11-24
> **Response Part [2/2]**
>
> ### **W3 & Q3: There appears to be no limitations section in this paper. What are the limitations?**
>
> We thank the reviewer for raising this point. In the revised manuscript,**&#x20;we have added a dedicated "Limitations" section in Appendix Sec. I**. This section explicitly discusses the main limitations of SWIRL:
>
> > To isolate the effect of the training scheme from rollout noise or environment stochasticity and to conduct controlled ablations, our experiments focus on static offline GUI datasets rather than dynamic online GUI environments.  In addition, our system infrastructure constraints prevented us from validating the method further in fully interactive settings.
> >
> > Moreover, SWIRL trades computation time for memory efficiency: the sequential update mechanism enables O(1) actor memory usage, but this design choice naturally increases the total wall-clock training time because each agent is optimized separately rather than concurrently.
> >
> > Finally, limited training resources prevented us from performing more extensive scaling experiments on substantially larger datasets, and a more systematic exploration of data scaling remains open for future investigation.
>
> ### **W4: The authors should be careful not to overreach in terms of their claims.**
>
> We appreciate the reviewer's careful reading. To clarify the scope of our claims and to better distinguish prior work on interleaved updates from the contributions specific to SWIRL, **we have revised the wording in several key locations**:
>
> > (1) Abstract (Around line 20)
> >
> > SWIRL instantiates a round-level training scheme that treats MARL as a sequence of single-agent reinforcement learning subproblems, updating one agent at a time while keeping the others fixed.
>
> > (2) Method description (Sec. 2.1)
> >
> > Alg. 1 provides a methodology that is conceptually related to prior alternating-update approaches in MARL (e.g., HAPPO, A2PO, MARFT), but differs in focusing on round-level inner solves and associated theoretical guarantees.

---

### Official Review · Reviewer_yu3B · 2025-11-02

**Soundness:** 3
**Presentation:** 3
**Contribution:** 3
**Rating:** 4
**Confidence:** 4

**Summary:**

SWIRL splits a GUI agent into a Navigator (plans) and an Interactor (executes) and trains them alternately, freeze one, optimize the other, so multi-agent training reduces to standard trust-region single-agent updates. This staged setup targets stability/efficiency and shows gains on mobile GUI control, with spillover benefits to math reasoning.

**Strengths:**

Clear role decomposition (auditable plan to act chain). Practical pipeline that reuses mature single-agent RL tooling. Theoretical footing (safety bound, round-wise monotonic improvement). Solid empirical wins for low-level execution and planner replacement. Transferable beyond GUI with modest data.

**Weaknesses:**

Training always freezes one agent (Navigator or Interactor) and optimizes the other, effectively decomposing the multi-agent problem into a sequence of single-agent GRPO/PPO/TRPO steps. This is closer to modular alternating optimization than classic concurrent cooperative MARL. As a result, non-stationarity and coordination that arise when both agents evolve simultaneously are not directly learned.

The GUI evaluation relies on offline single-step accuracy (Type/GR/SR)—e.g., a click landing in the box counts as correct, without reporting episode success rate, number of clicks, latency, or recovery/rollback behavior. High offline SR does not guarantee closed-loop, end-to-end performance.

The paper removes certain QA-style samples from GUI-Act-Web and evaluates on an older GUIOdyssey Test-Random split due to version changes. While this sharpens focus on manipulation skills, it alters the test distribution and complicates apples-to-apples comparison with prior work.

The reported stability stems from “alternating updates + online reweighting,” and ablations (e.g., removing reweighting) show clear degradation. This indicates robustness depends heavily on sample filtering and the training schedule, rather than fundamentally resolving non-stationarity under concurrent cooperative learning.

**Questions:**

see above

---

> ### Author Response · Authors · 2025-11-24
> **Response Part [1/3]**
>
> Dear Reviewer yu3B,
>
> Thank you for appreciating our paper's clear role decomposition, practical pipeline, theoretical grounding, strong empirical results, and transferability. Your constructive comments and suggestions are valuable to us. Below is our detailed response to address your concerns.
>
> ### **W1. Training is not classic concurrent cooperative MARL, so non-stationarity and coordination that arise when both agents evolve simultaneously are not directly learned.**
>
> Thank you for the careful reading and for raising this important point. We hope that the clarifications below more accurately position SWIRL as a round-level alternating training framework with both theoretical guarantees and empirical benefits, and that they address your concerns.
>
>
> **Clarifying the training scheme.** What we intent is to clarify that SWIRL is not a classic concurrently updated MARL algorithm. SWIRL is intentionally designed as a round-level alternating optimization scheme. During training, we update exactly one agent while freezing the other, thereby decomposing the joint multi-agent problem into a sequence of single-agent subproblems whose updates are interleaved to optimize the overall multi-agent objective. This design choice is deliberate, **aiming to retain multi-agent structure while enabling stable and tractable optimization in a resources-friendly approach.**
>
> **Stability and coordination results.** While SWIRL differs from concurrent MARL in its update mechanism, the alternating scheme **still enables stable learning dynamics and the emergence of coordination, as supported by both our theoretical analysis and empirical results.** In particular, **on the theoretical side (Sec 2.1)**, we establish a per-step safety lower bound, prove monotonic improvement across rounds, and show convergence of the return. **On the empirical side**, SWIRL yields both stable trainin&#x67;**&#x20;(Sec. F.2; Fig. 4)** and more effective co-evolution among agent&#x73;**&#x20;(Sec. F.1).**
>
> **Revision of ambiguous wording.&#x20;**&#x49;n the revised manuscript, we have carefully adjusted the description to more accurately reflect the alternating training nature of SWIRL. Specifically, the sentence in **Sec. 2.3:&#x20;**
>
> > "This approach is crafted to efficiently coordinate and enhance the performance of both the Navigator and Interactor concurrently."
>
> *&#x20;*&#x68;as been revised to:&#x20;
>
> > "This approach is crafted to efficiently coordinate and enhance the performance of the Navigator and Interactor through round-level alternating updates."
>
> *&#x20;*&#x57;e believe this revised wording more clearly conveys that SWIRL adopts a modular, round-based alternating optimization scheme rather than a concurrent cooperative MARL paradigm. We hope this clarification improves conceptual consistency and helps readers better understand the design rationale and training dynamics of our framework.

---

> ### Author Response · Authors · 2025-11-24
> **Response Part [2/3]**
>
> ### **&#x20;W2. The GUI evaluation relies on offline single-step accuracy**
>
> We thank the reviewer for raising this important point. We would like to clarify that offline GUI evaluation is a widely adopted and well-established paradigm in the GUI agent literature \[1–7]. Many recent works evaluate Type / GR / SR on static datasets because such settings enable rigorous, fine-grained comparisons across methods under exactly the same observations and ground-truth labels. For our paper, **this offline setting is particularly suitable, as our main contribution is the design and analysis of the SWIRL training** **mechanism**, and isolating the effect of the training scheme from rollout noise or environment stochasticity is essential for a clean evaluation.
>
> **Detailed Reason for offline evaluation.&#x20;**&#x4F;ur primary goal in this work is to propose and analyze the SWIRL framework for multi-agent training. To study the effect of the training scheme in a controlled manner, we deliberately follow standard offline protocols that fix the states, inputs, and supervision signals across different methods. As highlighted in prior work \[8], static closed-world datasets offer several advantages: they are lightweight to deploy, easy to reproduce, and allow fair, apples-to-apples comparison across algorithms without confounding factors from dynamic environments. By adopting Type / GR / SR on widely used GUI datasets, **we ensure that the performance differences we observe can be attributed to the learning framework itself rather than to environmental randomness or differences in rollout infrastructure.**
>
> **Limitations on online evaluation.** We agree that fully interactive, closed-loop online evaluation in GUI environments is an interesting and valuable direction for future work. However, **it is orthogonal to the central goal of this paper, which is to analyze the SWIRL training framework in a controlled setting where the effects of the training scheme can be cleanly isolated from environment-induced variance**. Building a large-scale online evaluation setup also requires substantial engineering effort and is beyond our current scope. We now clarify this practical constraint in **Appendix Sec. I** and highlight online evaluation as a natural next step enabled by our findings.
>
>
> \[1] AITW (Neurips 2023): https://arxiv.org/pdf/2307.10088
>
> \[2] AndroidControl (Neurips 2024): https://arxiv.org/pdf/2406.03679
>
> \[3] GUIOdyssey (ICCV 2025): https://arxiv.org/pdf/2406.08451
>
> \[4] Os-Atlas (ICLR 2024): https://arxiv.org/pdf/2410.23218
>
> \[5] CogAgent (CVPR 2024): https://arxiv.org/pdf/2312.08914
>
> \[6] UIPro (ICCV 2025): https://arxiv.org/pdf/2509.17328
>
> \[7] Mind2Web (Neurips 2023): https://arxiv.org/pdf/2306.06070
>
> \[8] GUI Agents: A Survey (ACL 2025): https://aclanthology.org/2025.findings-acl.1158.pdf

---

> ### Author Response · Authors · 2025-11-24
> **Response Part [3/3]**
>
> ### **W3. The paper removes certain QA-style samples from GUI-Act-Web and evaluates on an older GUIOdyssey Test-Random split due to version changes.&#x20;**
>
> **GUI-Act-Web.&#x20;**&#x41;s stated in the paper, the removed items are QA-style samples whose success mainly depends on open-ended web question answering rather than precise GUI manipulation. Our goal in this work is to measure manipulation competence, so we filter out these QA-style cases and keep only samples that directly test interaction with on-screen elements. We have re-evaluated the baselines on exactly the same filtered GUI-Act-Web split, as presented i&#x6E;**&#x20;Table A**. Under this matched setting, the performance of the baselines remains largely unchanged, while our model continues to achieve the strongest overall score. **We have updated Table 3 in the revised version to reflect these results.**
>
>
>
> Table A: Performance on GUI-Act-Web tasks after removing QA-style cases.
>
>
> | model      | TYPE        | GR        | SR        |
> | ---------- | ----------- | --------- | --------- |
> | UI-R1      | 80.69       | 80.07     | 67.92     |
> | GUI-R1-3B  | 94.07       | 86.86     | 73.53     |
> | GUI-R1-7B  | 93.15       | **88.33** | 76.15     |
> | Interactor | **95.00**   | 87.85     | **84.85** |
>
> **GUIOdyssey.&#x20;**&#x57;e believe there may have been a misunderstanding regarding the GUIOdyssey evaluation setup. Prior works (e.g., OS-Atlas, GUI-R1) report their results on the earlier release of GUIOdyssey and specifically on its Test-Random split. To enable a direct and fair apples-to-apples comparison with these baselines, we therefore follow the same protocol and also report results on this earlier Test-Random split. In other words, **all models (ours and all baselines) are evaluated on the same earlier version of GUIOdyssey, ensuring complete fairness in comparison.**
>
> ### **W4. Experiments indicate robustness depends on "alternating updates + online reweighting" together, rather than fundamentally resolving non-stationarity under concurrent cooperative learning.**
>
> Thank you for the careful reading and for highlighting the interaction between alternating updates and online reweighting in producing stable training behavior.
>
> **Clarifying the source of stability.** Rather than viewing alternating updates and reweighting as isolated "tricks", we emphasize that **they are integral components of the SWIRL framework**. Alternating updates control the optimization dynamics by holding one agent fixed, while reweighting adaptively shapes the learning signal to prioritize more reliable samples. Their combination forms a coherent training mechanism that promotes stability by design, not as a post-hoc patch. The ablations demonstrate that removing either component weakens the mechanism, which is consistent with the fact that both were intended to work together.
>
>
>
> **Positioning relative to concurrent cooperative MARL.** By design, SWIRL does not target the full resolution of non-stationarity in fully concurrent cooperative MARL. Instead, **it leverages a sequential multi-agent optimization scheme that moderates non-stationarity and structures the training dynamics.** Our stability claims are made within this setting, where the round-level alternation and adaptive reweighting operate jointly to produce predictable and reliable learning behavior. The empirical results reflect this mechanism functioning as designed.
>
> **Revising the wording for clarity.** To prevent potential misunderstanding that SWIRL claims to address non-stationarity in all MARL scenarios, we have updated the wording in **Appendix D.3** accordingly, as follows:
>
> > Fig. 4 shows consistent performance improvement across training phases. This stability arises from the overall SWIRL training mechanism, in particular from the combination of round-level alternating updates and online reweighting. Appendices F.1 and F.4 show that coordination improves over rounds, while Appendix F.3 demonstrates that allocating the same budget to more alternating rounds yields better results than deeper single-agent updates. While SWIRL does not aim to eliminate non-stationarity in fully concurrent MARL, it provides stable and theoretically grounded optimization in our sequential multi-agent framework.

---

### Author Response · Authors · 2025-11-24
**General Response: Thanks, Contributions, New Clarifications, and New Experiments**

We sincerely appreciate all reviewers for their time and thoughtful feedback. We are encouraged that the reviewers recognize the value of the proposed SWIRL framework for multi-agent training. From an engineering perspective, we reformulate the optimization problem of LLM-based multi-agent systems into a sequence of single-agent RL subproblems updated in an alternating manner. We further provide theoretical guarantees for this engineering design. As a result, SWIRL achieves stable training while maintaining an O(1) actor memory footprint, enabling effective multi-agent training even under limited computational resources. Our experiments demonstrate the effectiveness of the approach and its strong cross-domain transfer capabilities.

## **Contributions**
Main contributions recognized by reviewers are concluded as follows:
- **A resource-efficient framework for multi-agent training [yu3B, t83B, dLTb, myFk]:** We propose SWIRL, a staged workflow that optimizes multi-agent systems by alternately solving single-agent reinforcement learning subproblems, enabling the reuse of single-agent RL training tools while keeping the actor's memory footprint at O(1).
- **Theoretical guarantees [yu3B, t83B, dLTb, myFk]:** We establish a per-step safety bound, prove joint returns monotonic improvement, and show convergence of returns, thereby demonstrating that our method is not just an engineering heuristic but is backed by explicit theoretical guarantees.
- **Strong empirical results [yu3B, t83B, Qt5R, dLTb, myFk]:** We instantiate SWIRL for mobile GUI control with a Navigator and an Interactor, and through extensive experiments demonstrate stable training and state-of-the-art zero-shot performance, complemented by ablations that probe its underlying mechanisms.
- **Effective cross-domain transfer [yu3B, t83B, dLTb, myFk]:** We show that the same training recipe transfers effectively to the mathematics domain, yielding robust improvements on standard benchmarks and suggesting strong potential for broader multi-agent applications.

## **New Clarifications and New Experiments**
### **(1) New Clarifications**
- Provided a more detailed explanation of SWIRL to distinguish it from concurrent MARL in Sec. 1. [yu3B,Qt5R]
- Reorganized Sec. 2.1 to clarify the motivation and theoretical results. [myFk]
- Rigorously refined the presentation of the alternating update mechanism and its results in Sec. 1, 2.1, 2.3, and Appendix D.3. [t83B,yu3B]
- Included more valuable works in the related work (Appendix Sec. B). [myFk]
- Clarified how we adapt related theoretical results to derive our guarantees in Appendix C.2. [myFk]
- Introduced a dedicated limitations section in Appendix I. [t83B]

### **(2) New Experiments and Analysis**
We also thank all reviewers for their insightful and constructive suggestions, which helped a lot in further improving our paper. In addition to the pointwise responses below, we summarize supporting clarifications, experiments and analysis added in the rebuttal according to the reviewers' suggestions.

- Updated the result in Table 3. [yu3B]
- Instantiated a three-agent system in Appendix F.6. [t83B]
- Compared with the Isolated per-agent training in Sec. 3.5 and Table 7 [Qt5R]
- Conducted an in-isolation evaluation of the Navigator in Appendix F.7. [Qt5R]
- Incorporated additional offline evaluation comparisons on AitW in Appendix F.8. [myFk]
- Profiled the training wall-clock time in Appendix E.1.[t83B]
- Presented qualitative analysis illustrating how the Navigator and Interactor coordinate in Appendix H. [Qt5R]
- Elucidated qualitative examples illustrating enhanced mathematical reasoning performance in Appendix H. [dLTb]


We hope our pointwise responses below can clarify all reviewers' confusion and alleviate all concerns. Thanks to all reviewers, and **we have incorporated mentioned contents into the revised paper, highlighted in blue**.

---

### Comment · Area_Chair_MR3a · 2025-11-25

Dear Reviewers,

A quick reminder that the authors have posted their responses. The discussion period ends on December 2, so please review the rebuttal and share any follow-up comments as soon as possible. Your timely input is greatly appreciated. Thanks.

Best,

Your ACs

---

### Meta-Review · Area_Chair_UYuf · 2026-01-17

**Summary:**

Reviewers were mixed on the paper, initially leaning towards reject. They raised several main points:

1. Multi-agent vs single-agent framework and novelty. The reviewers mentioned that the framework is closer to alternating optimizing, and it's not clear while this needs to be a multi-agent framework. The reviewers also pointed out that this approach (navigator-interactor pipeline) is a common approach. These points were shared by 3 reviewers.

2. Reviewers were concerned by the evaluation and reliance on offline datasets, as well as single step accuracies without analyzing closed loop performance. They requested experiments on the wall clock times.

3. Reviewers had questions about baselines and requested additional experiments.

4. Reviewers had questions about the theory, and had questions about where the gains came from.

5. One reviewer was cocnerned about scalability beyond 2 agents.

**Reviewer Concerns:**

The authors did a good job at responding to many of the points. They added additional experiments and comparisons, including the wall clock time. They also clarified some of the questions about the theory.

The main outstanding points are the issues about a) novelty and the positioning with multi-turn agents, and b) the lack of closed loop evaluation and online episode success rates.

**Reviewer Scores:**

I expect the reviewers would have found many of the responses helpful, however the points about novelty and evaluation are major issues that the rebuttal did not convincingly address. These are key limitations of the paper, in particular the points about evaluation.

---

### Decision · Program_Chairs · 2026-01-26

Reject